# Intrinsic excitability mechanisms of neuronal ensemble formation

**Tzitzitlini Alejandre-García\*, Samuel Kim, Jesús Pérez-Ortega, Rafael Yuste**

Department of Biological Sciences, Columbia University, New York, United States

**Abstract** Neuronal ensembles are coactive groups of cortical neurons, found in spontaneous and evoked activity, that can mediate perception and behavior. To understand the mechanisms that lead to the formation of ensembles, we co-activated layer 2/3 pyramidal neurons in brain slices from mouse visual cortex, in animals of both sexes, replicating in vitro an optogenetic protocol to generate ensembles in vivo. Using whole-cell and perforated patch-clamp pair recordings we found that, after optogenetic or electrical stimulation, coactivated neurons increased their correlated activity, a hallmark of ensemble formation. Coactivated neurons showed small biphasic changes in presynaptic plasticity, with an initial depression followed by a potentiation after a recovery period. Optogenetic and electrical stimulation also induced significant increases in frequency and amplitude of spontaneous EPSPs, even after single-cell stimulation. In addition, we observed unexpected strong and persistent increases in neuronal excitability after stimulation, with increases in membrane resistance and reductions in spike threshold. A pharmacological agent that blocks changes in membrane resistance reverted this effect. These significant increases in excitability can explain the observed biphasic synaptic plasticity. We conclude that cell-intrinsic changes in excitability are involved in the formation of neuronal ensembles. We propose an 'iceberg' model, by which increased neuronal excitability makes subthreshold connections suprathreshold, enhancing the effect of already existing synapses, and generating a new neuronal ensemble.

**\*For correspondence:**
at3415@columbia.edu

## Editor's evaluation

This paper provides new insights regarding how the intrinsic excitability of neurons contributes to the formation of cortical ensembles, which may underlie memory formation. Previous work had left the relative contribution of intrinsic versus synaptic changes to neural ensemble formation incompletely understood. By using an interdisciplinary approach, the authors reveal the mechanisms by which changes in intrinsic excitability may play a unique role in the formation of new neural ensembles.

## Introduction

The function of the brain is anchored on the activity of groups of connected neurons, forming microcircuits (*Schüz and Braitenberg, 2001*; *Shepherd, 2004*). In such circuits, groups of coactive neurons, or ensembles (also known as assemblies, synfire chains, or attractors), found in both spontaneous and evoked activity, exhibit synchronous or correlated activity (*Abeles, 1991*; *Buzsáki, 2010*; *Carrillo-Reid et al., 2016*; *Cossart et al., 2003*; *Morris, 1999*; *Hopfield, 1982*; *Lorente De Nó, 1938*; *Marshel et al., 2019*; *Miller et al., 2014*; *Yuste, 2015*) and could mediate memory formation (*Morris, 1999*), and functional brain states (*Hoshiba et al., 2017*). In the cerebral cortex, ensembles could arise from reverberating patterns of multineural activity, generated by recurrent connectivity, and occur spontaneously (*Churchland and Sejnowski, 1992*; *Lorente De Nó, 1933*). In mouse visual cortex, cortical ensembles are causally related to memory storage and percepts (*Carrillo-Reid et al., 2019*; *Marshel*

**eLife digest** In the brain, groups of neurons that are activated together – also known as neuronal ensembles – are the basic units that underpin perception and behavior. Yet, exactly how these coactive circuits are established remains under investigation.

In 1949, Canadian psychologist Donald Hebb proposed that, when brains learn something new, the neurons which are activated together connect to form ensembles, and their connections become stronger each time this specific piece of knowledge is recalled. This idea that 'neurons that fire together, wire together' can explain how memories are acquired and recalled, by strengthening their wiring.

However, recent studies have questioned whether strengthening connections is the only mechanism by which neural ensembles can be created. Changes in the excitability of neurons (how easily they are to fire and become activated) may also play a role. In other words, ensembles could emerge because certain neurons become more excitable and fire more readily.

To solve this conundrum, Alejandre-García et al. examined both hypotheses in the same system. Neurons in slices of the mouse visual cortex were stimulated electrically or optically, via a technique that controls neural activity with light. The activity of individual neurons and their connections was then measured with electrodes.

Spontaneous activity among connected neurons increased after stimulation, indicative of the formation of neuronal ensembles. Connected neurons also showed small changes in the strength of their connections, which first decreased and then rebounded after an initial recovery period.

Intriguingly, cells also showed unexpected strong and persistent increases in neuronal excitability after stimulation, such that neurons fired more readily to the same stimulus. In other words, neurons maintained a cellular memory of having been stimulated. The authors conclude that ensembles form because connected neurons become more excitable, which in turn, may strengthen connections of the circuit at a later stage.

These results provide fresh insights about the neural circuits underpinning learning and memory. In time, the findings could also help to understand disorders such as Alzheimer's disease and schizophrenia, which are characterised by memory impairments and disordered thinking.

*et al., 2019*; *Miller et al., 2014*). Moreover, disorganized neuronal ensembles are observed in mouse models of schizophrenia (*Hamm et al., 2017*).

A hypothesis of how a neuronal ensemble (i.e. a neuronal assembly) could be formed was initially proposed by Hebb in 1949. Ensembles could be formed by synaptic plasticity among coactive neurons, as a consequence of the Hebbian rule, whereby the "persistent and repeated activation of connected neurons induce metabolic changes or growth of processes in one or both cells that strengthen those connections". One synaptic mechanism that fulfills Hebb's rule is long-term potentiation (LTP), a long-lasting increase in synaptic efficacy after a high frequency burst electrical stimulation, which is postulated to be the cellular correlate of learning (*Bliss and Gardner-Medwin, 1973*). With LTP, increased synaptic weights enhances preferential connectivity that leads to an increased recurrent activity within ensembles (*Hoshiba et al., 2017*). Consistent with this, synchronous stimulation of groups of neurons 50–100 times in vivo, using two-photon optogenetics, bound them together into an 'imprinted' ensemble that became spontaneously coactive after the optogenetic imprinting (*Carrillo-Reid et al., 2016*). However, no direct evidence has yet demonstrated that the formation of ensembles depends on changes in synaptic connections.

As an alternative hypothesis, recent experiments have revealed widespread activity-dependent mechanisms in cortical neurons, suggesting that the formation and stability of an ensemble, or a memory engram, could be due to cell-autonomous intrinsic mechanisms, such as changes in neuronal excitability (*Debanne and Poo, 2010*; *Gallistel and Balsam, 2014*; *Hansel and Disterhoft, 2020*; *Lisman et al., 2018*; *Titley et al., 2017*). This hypothesis is consistent with experiments where specific patterns of activity in the hippocampus recalls stored memories, under protein synthesis inhibitors, which should prevent LTP (*Pignatelli et al., 2019*; *Ryan et al., 2015*; *Tonegawa et al., 2015*). Moreover, individual Purkinje cells can acquire and represent temporal pattern of inputs in a cell intrinsic manner, without synaptic plasticity (*Johansson et al., 2014*). Thus, ensemble formation could be due

to intrinsic changes in the excitability state of the neurons, instead of, or in addition to, strengthening of its synapses (*Disterhoft and Oh, 2006*; *Ganguly et al., 2000*; *Pignatelli et al., 2019*).

To explore mechanisms of ensemble formation, we optogenetic and electrically stimulated neurons in mouse neocortical slices, replicating in vitro the optogenetic protocol used to imprint ensembles in vivo (*Carrillo-Reid et al., 2016*). We then used whole-cell and perforated patch recordings, from individual neurons and connected pairs, to electrophysiologically characterize the stimulated neurons. We find moderate biphasic changes in synaptic strength, in the midst of major and generalized increases in cell-intrinsic excitability, that can, at least partly, explain the synaptic changes observed. We conclude that changes in excitability play a major role in ensemble formation.

## Results

### Spontaneous activity becomes correlated in simultaneously stimulated neurons

A neuronal ensemble can be defined as a group of neurons with correlated spontaneous and evoked activity (*Buzsáki, 2010*; *Cossart et al., 2003*; *Ikegaya et al., 2004*; *Sasaki et al., 2007*; *Shepherd and Grillner, 2010*; *Stringer et al., 2019*; *Yuste, 2015*). Previous results have demonstrated that optogenetic coactivation of neurons can generate new ensembles in vivo (*Carrillo-Reid et al., 2016*). To explore the cellular and circuit mechanisms underlying this phenomenon, we studied the electrophysiological mechanisms that led to building of imprinted ensembles, replicating in vitro with the activation protocols that generates ensembles in vivo, with optogenetic or electrical stimulation. Experiments were performed in layer 2/3 pyramidal neurons expressing the opsin ChroME in slices from primary visual cortex of adult mice of both sexes (*Figure 1A*).

Optogenetic imprinting in vivo increased the correlation of spontaneous activity of stimulated neurons, indicative of the formation of a new ensemble (*Carrillo-Reid et al., 2016*). To explore if our in vitro stimulation protocols were effective in generating ensembles, we used paired recordings to explore whether simultaneous and repeated optogenetic or electrical activation neurons also induced changes in the correlation of spontaneous activity. We recorded pairs of closely located pyramidal neurons and found that 25% of recorded pairs, at distances of 20–30 μm, were monosynaptically connected (7/28 pairs), consistent with previous reports (*Cossell et al., 2015*; *Holmgren et al., 2003*; *Ko et al., 2011*; *Lefort et al., 2009*; *Levy and Reyes, 2012*; *Markram et al., 1997*; *Song et al., 2005*; *Figure 1A–C*). Also, in agreement with previous studies, we found higher values of correlation of spontaneous activity between connected neuronal pairs (0.31±0.16; n=7) than between unconnected neuronal pairs (–0.04±0.37; n=21; p=0.049 by Mann-Whitney test, *Figure 1F*; *Cossell et al., 2015*; *Ko et al., 2011*; *Yoshimura et al., 2005*). However, ~50% (10/21) of the unconnected neuronal pairs still had relatively high correlated activity, comparable to connected neuronal pairs (0.32±0.18; n=9; all unconnected pairs with correlation >zero; *Figure 1B*). Notably, after optogenetic or electrical stimulation (*Figure 1D and E*), the correlation of spontaneous activity between neurons increased significantly, in both connected and unconnected pairs (0.02±0.06–0.2±0.05; n=13 pairs, 2 connected, 11 unconnected; p=0.049 by Wilcoxon; *Figure 1G*). No significant differences in correlations were found in unstimulated pairs (0.05±0.1–0.03±0.1, n=11 pairs, p=0.7 Wilcoxon, *Figure 1H*). We conclude that optogenetic and electrical stimulation can increase the correlation of spontaneous activity among neurons, a hallmark of neuronal ensemble formation.

### Effect of optogenetic stimulation on monosynaptic currents

The creation of ensembles in vivo after optogenetic stimulation (*Carrillo-Reid et al., 2016*), suggested that ensembles could be built by Hebbian synaptic plasticity, by potentiation of existing synapses of coactivated neurons. To explore this, we performed dual whole-cell and perforated patch-clamp recordings of connected neurons, and examined whether persistent and repeated optogenetic or electrical activation induced synaptic plasticity (*Cossell et al., 2015*; *Morris, 1999*). Action potentials were induced in the presynaptic neuron with depolarizing currents (10 pulses; 400–600 pA, 5ms, 20 Hz trains). For measuring synaptic currents, postsynaptic neurons were recorded in voltage-clamp, without current injection. We classified connections as monosynaptic if EPSC latency was 2±1 ms, relative to the presynaptic spike (*Figure 2A*; n=7). To normalize EPSC amplitudes, postsynaptic current values in the action potential start time were considered baseline. Current values for the first EPSC

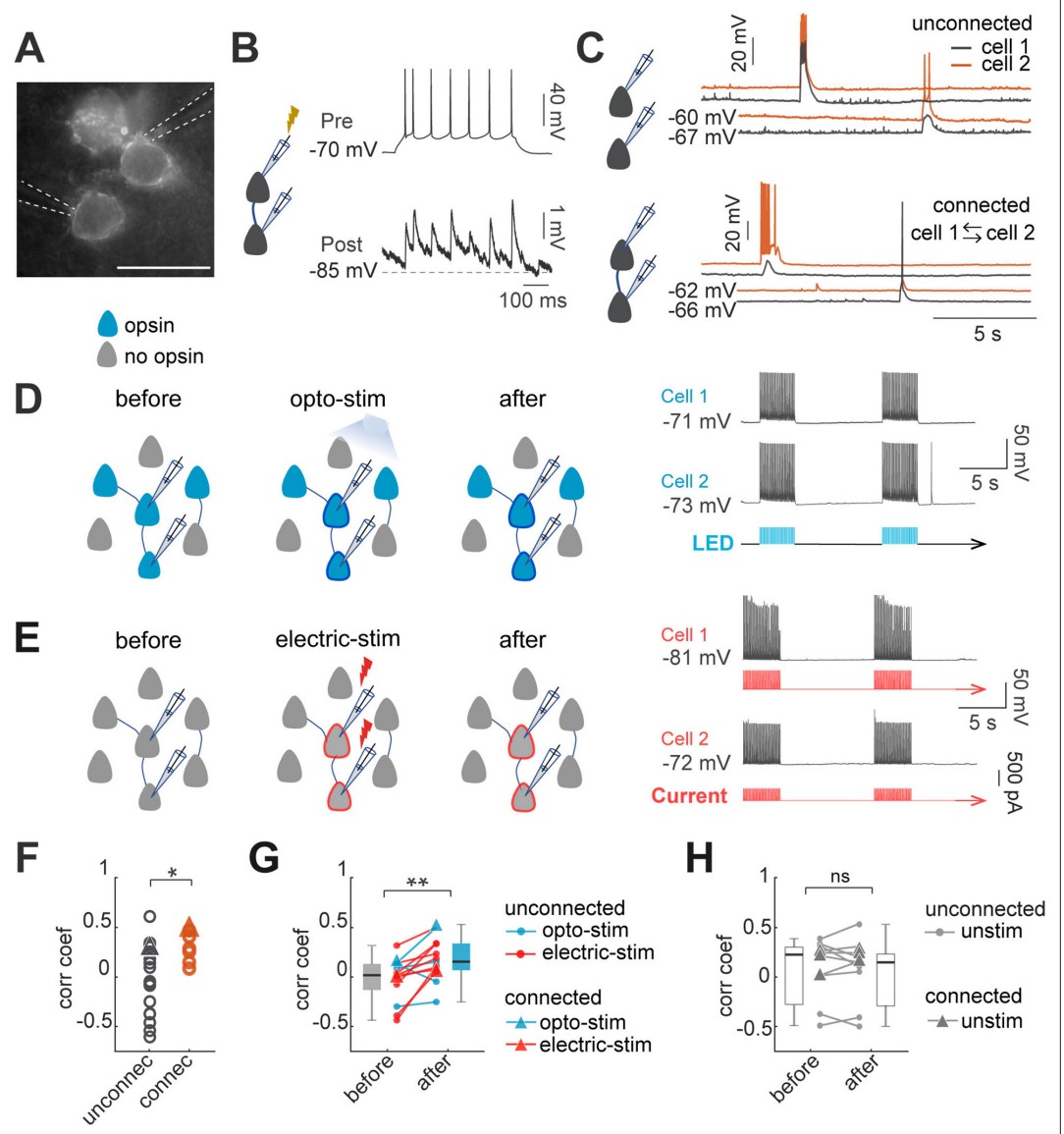

**Figure 1.** Optogenetic and electrical co-stimulation increases correlations of spontaneous activity. (**A**) Image of Ruby3 reporter fluorescent of ST-ChroMe opsin expression in L2/3 pyramidal neurons in primary visual cortex, in a brain slice of an in-utero electroporated mice. Scale bar: 40 μM. (**B**) Paired recording measurements of monosynaptic connectivity. Perforated patch-clamp recording of presynaptic action potentials elicited by 100 pA current injections (500ms), followed by identification of a monosynaptic connection, generating postsynaptic potentials, time-locked to presynaptic spikes. (**C**) Spontaneous activity of two pairs. Top: current-clamp spontaneous activity of unconnected neurons (correlation coefficient = 0.33). Bottom: Representative paired recording from neurons that were bi-directionally connected (correlation coefficient = 0.49). (**D**) Experimental design of optogenetic protocol. Perforated current-clamp recording of two neurons with opsin expression in blue: 1–30 min of 10 Hz train, 5ms light pulses for 4 s followed by 10 s of rest. (**E**) Experimental design of electrical stimulation protocol. Perforated current-clamp recording of two neurons without opsin expression in gray: 1–30 min of 10 Hz train, 5ms current pulses for 4 s followed by 10 s of rest. (**F**) Correlation coefficients of connected and unconnected neurons. Correlation coefficients were calculated for the first 3 min of simultaneous recording in each pair; triangles represent correlation values of pairs showed in C. Correlation coefficients of connected (n=7 pairs) vs. unconnected (n=21 pairs) showed significant differences p=0.049 by Mann Whitney; 15 mice. (**G**) Correlation between pairs increased after continued co-stimulation. The graph shows an increase in the correlation coefficient before and after 20–30 min of continuous optogenetic (blue) or electrical stimulation (red) (p=0.004 by Wilcoxon test; n=13 pairs, 12 mice). (**H**) Correlation before and after 20–30 min without stimulation (gray) (p=0.7 by Wilcoxon; n=11 pairs, 8 mice). Triangles represent synaptically connected pairs.

*Figure 1 continued on next page*

*Figure 1 continued*

The online version of this article includes the following source data for figure 1:

**Source data 1.** Correlation coefficients of connected and unconnected neurons.

**Source data 2.** Correlation coefficients of pair of neurons before and after optogenetic or electrical stimulation.

(EPSC$_1$) during the stimulation trains, including failures, were averaged. A connection was defined as existent if the average evoked EPSC$_1$ had an amplitude larger than 1.5 S.D. of the noise. Control recordings were performed with unconnected neurons (n=21 pairs), and the optogenetic protocol did no generate any new detectable synaptic connections.

In pairs of connected neurons, we examined if optogenetic stimulation induced changes in monosynaptic currents, pooling data from whole-cell (n=5) and perforated patch-clamp (n=2) recordings. The average amplitude of monosynaptic ESPCs in control condition agreed with previous reports (*Feldmeyer et al., 2006*; *Morishima et al., 2011*; *Sempere-Ferràndez et al., 2019*; *Yoshimura et al., 2005*). Although average EPSC$_1$ peak currents of optogenetically stimulated neurons did not show significant changes with stimulation, they revealed a trend towards amplitude depression after the stimulation, followed with a tendency towards potentiation, after a 20 min rest period (before: 13±7 pA; after: 9±7 pA; p=0.2 Wilcoxon; n=7, *Figure 2B and C*). Control EPSC$_1$ amplitudes of unstimulated neurons did not show significant difference before (13±2 pA) and after 30 min of recordings (12±2 pA; p=0.1 Wilcoxon; n=11, whole-cell recordings, n=7; perforated patch-clamp, n=4). To evaluate changes in short-term plasticity, we calculated the paired-pulse ratio (PPR), that is the ratio of amplitudes of the second to the first EPSC in the train (EPSC$_2$/EPSC$_1$). We integrated data from whole-cell (n=5) and perforated patch-clamp recordings (n=2), and found that PPR did not significantly change with the stimulation (before: 0.95±0.09 PPR; after: 0.7±0.1 PPR; p=0.4 Wilcoxon; n=7; *Figure 2D*). Similarly, control unstimulated neurons in whole-cell (n=4) and perforated patch-clamp recordings (n=7) did not show significant changes in PPR before (1.08±0.12) and after 30 min (1.06±0.12; p=0.7 Wilcoxon; n=11).

We concluded that optogenetic stimulation did not have a significant effect on monosynaptic currents, although with a consistent trend towards an initial depression, followed with a subsequent potentiation.

## Biphasic plasticity of monosynaptic currents after electrical stimulation

We carried out similar experiments with electrical stimulation of connected pairs of neurons, thus avoiding possible side effects caused by opsin expression (see Materials and methods) or the optogenetic stimulation of other neurons expressing opsin in the field of view. Using perforated patch-clamp, we recorded and electrically stimulated pairs of connected pyramidal neurons. After electrical stimulation, EPSC$_1$ amplitudes decreased significantly (from 14±2 pA to 8±1 pA; p=0.016 by Wilcoxon; n=7; *Figure 2E*). After pausing the stimulation for 20 min, the amplitude of evoked ESPCs recovered, with a moderate potentiation. Specifically, for electrically stimulated neurons, EPSC$_1$ amplitude recovered significantly from 8±1–20±1 pA after 20 min of post-stimulation rest (p=0.015 by Wilcoxon; n=7). The amplitude after the recovery period was moderately larger than the EPSCs amplitude before stimulation (14±2–20±1 pA; p=0.015 by Wilcoxon; *Figure 2E*). In contrast, control unstimulated neurons did not show a significant change (before: 13±2 pA; after 30 min: 12±2 pA; p=0.6 by Wilcoxon; n=8). We also measured the PPR of electrical stimulated neurons before and after electrical stimulation, finding no significant differences (before: 1.3±0.15 PPR; after: 1.03±0.18 PPR; p=0.7 Wilcoxon; n=7). No difference in PPR was observed before and after 20 min post-stimulation rest (1.3±0.13; p=0.1 by Wilcoxon; *Figure 2E*). Similarly, the PPR of unstimulated neurons did not change before (1.4±0.1) and after 30 min without stimulation (1.3±0.15; p=0.5 by Wilcoxon; n=7) (*Figure 2F*).

We conclude that electrical coactivation leads to a moderate biphasic change in monosynaptic currents, with an initial depression followed by a potentiation. These results are consistent with the trends found in pairs of optogenetic stimulated neurons, where EPSC$_1$ amplitude after a post-stimulation recovery period increased from 9±7 pA to 19±3 pA, although no statistical comparison was possible due to the low n (*Figure 2C*). The difficulty in obtaining perforated whole-cell recordings from connected pairs of neurons in opsin expressing cells from adult brain slices precluded us from

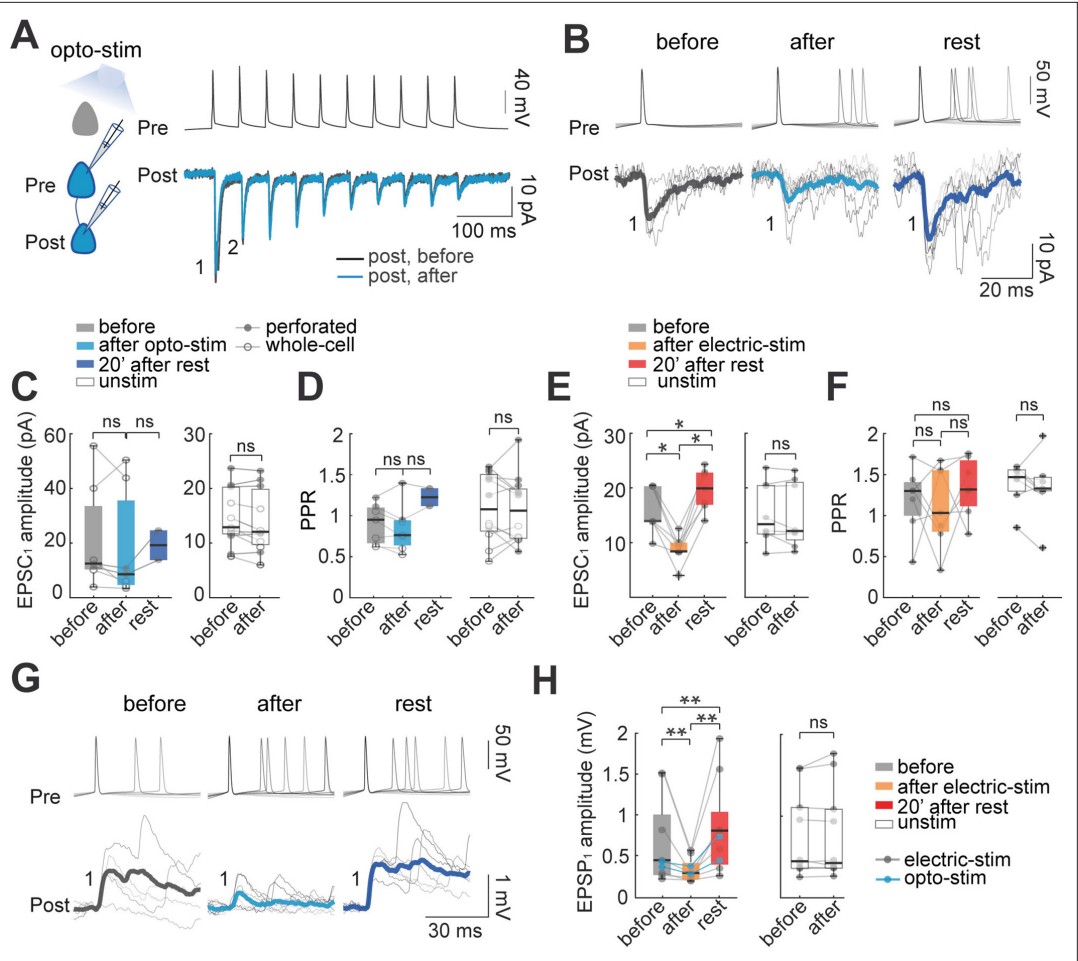

**Figure 2.** Effect of optogenetic and electrical stimulation in monosynaptic EPSCs and EPSPs. (**A**) Representative paired whole-cell recording of synaptically connected neurons. Top: current-clamp recording of presynaptic action potentials in response to 10 current injections (5ms each; 20 Hz). Bottom: voltage-clamp recording of evoked EPSC before (black) and after (blue) 30 min of optogenetic stimulation. Each trace is average of 30 successive responses evoked by presynaptic current injection. (**B**) Representative paired recording of evoked EPSCs (perforated patch-clamp). Top: current-clamp recording of presynaptic action potentials induced by positive current steps of the I-V curve (20–120 pA). Bottom: voltage-clamp recording of evoked EPSCs before and after optogenetic stimulation and after 20 min of rest post-stimulation. Thick line is average of successive responses to the first presynaptic action potentials, for every current step of the I-V curve. (**C**) $EPSC_1$ amplitudes without significant changes before and after optogenetic stimulation. Left: mean $EPSC_1$ amplitude before and after optogenetic stimulation. Open dots correspond to whole-cell recordings, n=5; filled dots correspond to perforated patch-clamp recordings; n=2. Medians and blue box plot: p=0.2 by Wilcoxon; n=7 neurons, 5 mice. Right: Unstimulated neurons, n=7 neurons with whole-cell and n=4 neurons with perforated patch-clamp. Medians and white box plots; p=0.1 by Wilcoxon; n=11 postsynaptic neurons, 8 mice. (**D**) Comparison of mean Paired Pulse Ratio (PPR). Left: Mean PPRs before and after optogenetic stimulation and after 20 min post-stimulation. PPR show no significant differences between before and after, p=0.4; Wilcoxon; n=7 cells, 5 mice. Right: PPR of unstimulated neurons did not show significant differences, p=0.6 by Wilcoxon, n=11 cells, 8 mice. (**E**) The amplitude of evoked $EPSC_1$ after electrical stimulation. Comparable to optogenetic stimulation, electrical stimulation protocol consisted of 30 min of 10 Hz train, 5ms current pulses for 4 s followed by 10 s of rest in two neurons simultaneously, this protocol was done in brain slices without opsin expression and using only perforated patch-clamp. Left: the average $EPSC_1$ amplitude decreased after electrical stimulation (p=0.015 by Wilcoxon; n=7 postsynaptic neurons, 5 mice). $EPSC_1$ amplitudes recovered with a moderate increase, after 20 min of rest post-stimulation (p=0.015), and also compared to the period before stimulation (p=0.015). Right: $EPSC_1$ amplitude of unstimulated neurons (p=0.7 by Wilcoxon; n=8 neurons, 5 mice). (**F**) PPR remains unchanged: before and after electrical stimulation (p=0.7 Wilcoxon; n=8 cells, 5 mice); also, after electrical stimulation and rest post-stimulation (p=0.2); comparison before and rest post-stimulation did not show differences either (p=0.1), as well as PPR of unstimulated neurons; p=0.6 by Wilcoxon, n=12 cells, 8 mice.

*Figure 2 continued on next page*

*Figure 2 continued*

(**G**) Representative paired recording of evoked EPSPs (perforated patch-clamp). Top: current-clamp recording of presynaptic action potentials. Bottom: current-clamp recording of evoked EPSPs before, after optogenetic stimulation and, after 20 min of rest post-stimulation. Thicker lines are average of successive responses to presynaptic action potentials. (**H**) Comparison of mean $EPSP_1$ amplitude after optogenetic stimulation (blue dots; n=2) and after electrical stimulation (gray dots; n=7). Left: average $EPSP_1$ amplitude decreased after stimulation (p=0.004 by Wilcoxon; n=9 neurons, 5 mice). $EPSP_1$ amplitudes recovered with a moderate increase after 20 min of post-stimulation (p=0.004 by Wilcoxon) and also compared to before stimulation (p=0.008 by Wilcoxon). Right: $EPSP_1$ amplitude of unstimulated neurons (p=0.7 by Wilcoxon; n=10 postsynaptic neurons, 5 mice).

The online version of this article includes the following source data for figure 2:

**Source data 1.** Evoked unitary EPSC before and after optogenetic stimulation.

**Source data 2.** Evoked unitary EPSC before and after electrical stimulation.

**Source data 3.** Evoked unitary EPSP before and after optogenetic and electrical stimulation.

---

increasing the n after many attempts. These difficulties were ameliorated in the electrical stimulation experiments.

## Biphasic plasticity of synaptic potentials after stimulation

To further explore potential postsynaptic mechanisms of ensemble formation, in parallel experiments, we used current-clamp to examine postsynaptic potentials (EPSPs) in connected pairs, comparing evoked EPSP amplitudes before and after optogenetic and electrical stimulation (*Figure 2G*). We combined data from optogenetically (n=2, blue) and electrically-stimulated pairs (n=7, gray). In good agreement with the voltage-clamp data, the average peak amplitude of the first EPSP ($EPSP_1$) significantly decreased after stimulation (0.43±0.2 mV to 0.28±0.2 mV; p=0.004 by Wilcoxon; n=9; *Figure 2H*). As with EPSC measurements, EPSPs recovered and became potentiated after 20 min post-stimulation to 0.8±0.2 mV (recovery compared with before: p=0.008; and after stimulation: p=0.004 by Wilcoxon). Control unstimulated neurons did not show significant differences in EPSP amplitude before (0.44±0.2 mV) or after 30 min (0.42±0.2 mV; p=0.7 by Wilcoxon; n=10).

In conclusion, paired recordings, in both current and voltage clamp, revealed that the coactivation of neurons generates a moderate biphasic synaptic plasticity, with an initial small depression, followed by a rebound potentiation, after several minutes of rest.

---

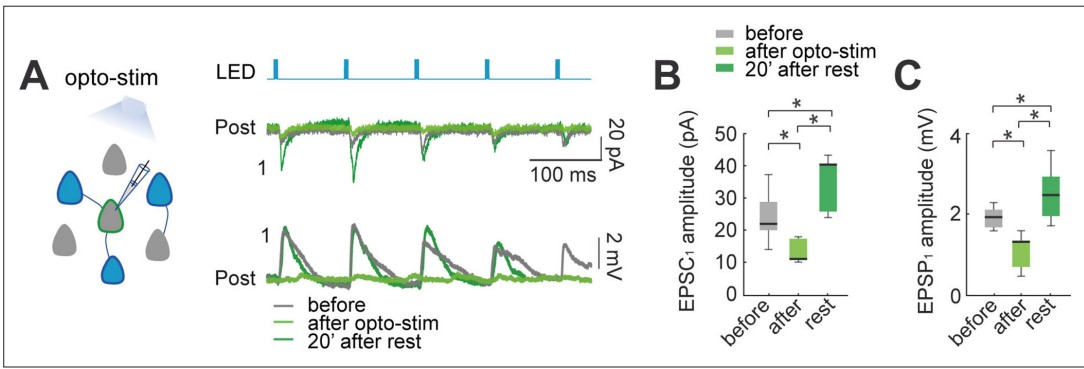

**Figure 3.** Effect of optogenetic stimulation in synaptic plasticity of the local circuit. (**A**) Activation of opsin expressing neuronal population in the slice by LED pulses generates evoked EPSCs and EPSPs in non-expressing pyramidal neurons. Voltage-clamp recording (top) and current-clamp recording (bottom) of a representative non-expressing neuron. Evoked EPSCs and EPSPs before (gray); after 30 min of optogenetic stimulation (green), and 20 min post stimulation (dark green). (**B**) The amplitude of population light-evoked $EPSC_1$ (left) and (**C**) light-evoked $EPSP_1$ (right) decreased after 30 min of optogenetic stimulation (p=0.01 and p=0.01 respectively, by Wilcoxon; n=7 postsynaptic neurons; 7 mice), and recovered with a moderate increase after 20 min of rest (p=0.01 and p=0.01, respectively).

The online version of this article includes the following source data for figure 3:

**Source data 1.** Evoked local circuit EPSP and EPSC before and after optogenetic stimulation.

## Biphasic synaptic plasticity in local circuit after optogenetic stimulation

To explore synaptic plasticity in the local circuit of optogenetically stimulated neurons, we also performed voltage-clamp and current-clamp recordings of evoked EPSCs and EPSPs in neurons *without* opsin expression, but located in the same field of view (*Figure 3A*). Assuming an area photo-stimulated by the LED of 150 µM in radius, we estimated that the number of superficial neurons with opsin expression potentially activated by the light was ~20 cells. Thus, optical stimulation could lead to an increase in the activity of the local circuit. To monitor this, we used perforated patch-clamp from non-expressing pyramidal neurons, and measured EPSCs or EPSPs evoked by the LED, in both voltage and current-clamp. Light-evoked EPSCs or EPSPs were analyzed if the latency was >2ms relative to LED pulse onset (*Figure 3A*; n=7). In non-expressing cells, population light-evoked $EPSC_1$ (22±3 pA) and $EPSP_1$ amplitudes (2±0.1 mV) were significantly higher than unitary evoked $EPSC_1$ (14±2 pA; p=0.04 by Mann-Whitney) or $EPSP_1$ (0.43±0.2 mV; p=0.0006 by Mann-Whitney) found in connected pairs. This is likely because LED stimulated several neurons in the field simultaneously, generating a compound EPSC and EPSP.

After optogenetic stimulation (10 Hz of 5ms pulse for 4 s every 10 s; 30 min), the average light-evoked $EPSC_1$ showed a significant diminution in amplitude (22±3–11±1 pA; p=0.015 by Wilcoxon; n=7; *Figure 3B*), and a recovery with potentiation after 20 min post-stimulation (40±3 pA; p=0.015, comparing with before; p=0.015, Wilcoxon). Similarly, light-evoked EPSPs showed a significant decrease after 30 min optogenetic stimulation (2±0.1 mV to 1.3±0.2 mV; p=0.015 by Wilcoxon; n=7; *Figure 3C*), and also recovered with a moderate increase in amplitude after 20 min post-stimulation (2.5±0.3 mV; p=0.015, comparing with before; p=0.015, Wilcoxon). These results, in excellent agreement with the connected pair data, reveals that the biphasic synaptic plasticity is also found in unstimulated neurons, thus suggesting that it is a circuit-wide phenomenon.

## Increased frequency and amplitude of spontaneous synaptic inputs after optogenetic and electrical stimulation

To further explore the effect of the optogenetic and electrical stimulation on population activity, we measured spontaneous EPSPs from recorded neurons, and compared them before and after stimulation (*Figure 4A*). To measure spontaneous EPSPs, we low-pass filtered current-clamp recordings, subtracting membrane potential oscillations, to obtain comparable baselines of EPSPs amplitudes (*Juárez-Vidales et al., 2021*). We detected EPSPs with amplitudes between 0.3 mV and 10 mV and analyzed changes in the frequency of spontaneous EPSPs before and after optogenetic stimulation (*Figure 4A–B*). The frequency of spontaneous EPSPs significant increased from 90±9 EPSPs/min before to 127±19 after optogenetic stimulation (p=0.004 by Wilcoxon; n=10) (*Figure 4D* left). We also observed increases in EPSPs amplitude (0.38±0.01–0.41±0.02 mV; p=0.04) (*Figure 4E* left). In control recordings (*Figure 4D–E* right), non-expressing pyramidal neurons showed no statistically differences in the frequency of spontaneous EPSPs (93±25–64±18; p=0.06), or in EPSPs amplitude (0.37±0.004–0.37±0.004 mV; p=0.5; n=7).

Similarly, electrical stimulated neurons showed a significant increase in the frequency of spontaneous EPSPs from 80±12–118±17 EPSPs/min (p=0.001 by Wilcoxon; n=13) and EPSPs amplitude (0.37±0.03–0.40±0.03 mV; p=0.03) (*Figure 3F–G* left). In unstimulated neurons (*Figure 3F–G* right), we found no statistically difference in the frequency of spontaneous EPSPs (98±18–93±15; p=0.4), or any significant differences in EPSPs amplitude (0.37±0.01–0.38±0.01 mV; p=0.4; n=12).

These results reveal significant changes in spontaneous synaptic inputs after optogenetic and electrical stimulation. The increase in spontaneous EPSPs amplitude could partly explain their increased frequencies, as smaller EPSPs will become detectable. Importantly, since increases in spontaneous EPSPs frequency and amplitude also occur after electrical stimulation of individual neurons, our results demonstrate that neurons become more sensitive to synaptic inputs after stimulation, in a cell intrinsic manner.

## Optogenetic stimulation increases neuronal excitability

Our results demonstrated that optogenetic and electrical stimulation, even of individual neurons, could induce changes in intrinsic cellular excitability. This is in agreement with recent literature that has described non-synaptic cellular plasticity in a variety of experimental preparations (*Abraham et al., 2019*; *Campanac et al., 2008*; *Ganguly et al., 2000*; *Malik and Chattarji, 2012*; *Marder and*

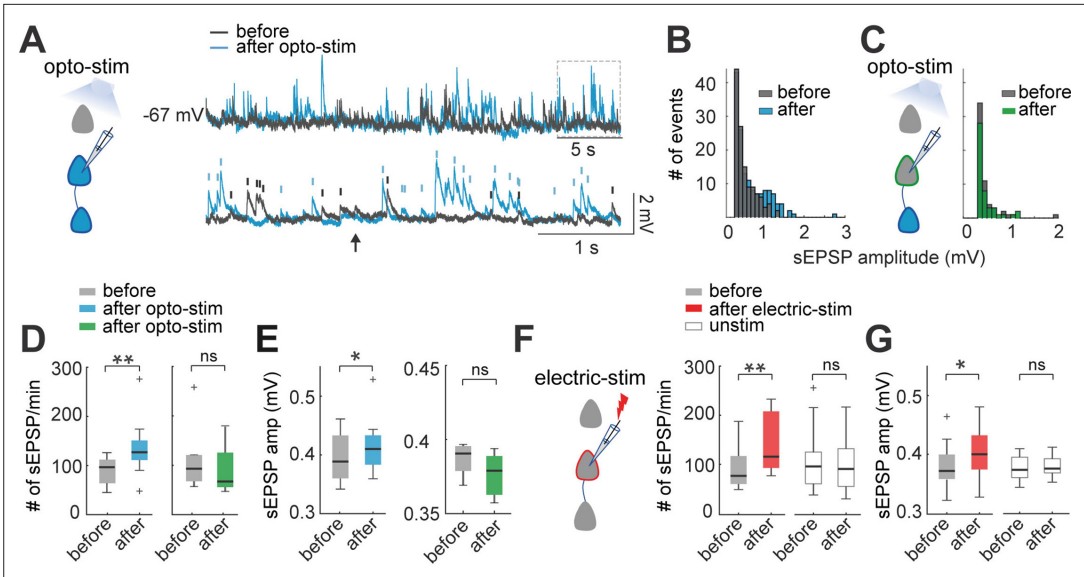

**Figure 4.** Optogenetic and electrical stimulation increases frequency and amplitudes of spontaneous EPSPs.
(**A**) Representative perforated patch-clamp recording of a neuron in current-clamp. Top: spontaneous EPSPs
of a neuron before and after optogenetic stimulation. Bottom: Section of top trace shows spontaneous EPSPs
amplitude >0.3 mV before (gray) and after (blue) optogenetic stimulation. Arrow shows a putative EPSP <0.3 mV
below threshold (0.3 mV). (**B**) Frequency histogram of spontaneous EPSPs amplitudes shows that, after optogenetic
stimulation, the number of events increased, as well as the number of events with higher amplitudes. (**C**) Frequency
histogram of spontaneous EPSPs amplitudes shows that the number of events of a non-expressing pyramidal
neuron remained unchanged before and after optogenetic stimulation of expressing neuronal population. (**D**) The
number of spontaneous EPSPs increased in optogenetically stimulated neurons but not in non-expressing neurons.
Left: Optogenetic stimulation of neurons: blue median and box plot; p=0.004 by Wilcoxon; 8 mice, n=10. Right:
Number of spontaneous EPSPs of non-expressing pyramidal neurons before and after optogenetic stimulation
protocol: green median and box plot; p=0.06 by Wilcoxon; 5 mice, n=7 (**E**) The amplitude of spontaneous EPSPs
increased in optogenetically stimulated neurons but not in non-expressing neurons. Right: Optogenetic stimulated
neurons: blue median and box plot; p=0.04 by Wilcoxon. Left: Non-expressing neurons: green median and box
plot; p=0.5 by Wilcoxon. (**F**) The number of spontaneous EPSPs increased after electrical stimulation. Electrical
stimulated neurons: red medians and box plot; p=0.0012 by Wilcoxon; 9 mice, n=13. Unstimulated neurons: white
medians and box plot; p=0.4 by Wilcoxon; n=12; 9 mice. (**G**) The amplitude of spontaneous EPSPs increased
after electrical stimulation, n=13. Electrical stimulated neurons: red median and box plot; p=0.003 by Wilcoxon.
Unstimulated neurons: white medians and box plots; p=0.4 by Wilcoxon.

The online version of this article includes the following source data for figure 4:

**Source data 1.** Spontaneous EPSPs before and after optogenetic stimulation.

**Source data 2.** Spontaneous EPSPs before and after electrical stimulation.

*Goaillard, 2006*; *Paz et al., 2009*; *Ryan et al., 2015*; *Titley et al., 2017*; *Xu et al., 2005*; *Yang and
Santamaria, 2016*). To explore these mechanisms, we evaluated active and passive intrinsic elec-
trophysiological parameters, such as current injection-dependent firing, frequency, membrane resis-
tance, membrane potential, and firing threshold. We recorded individual neurons with perforated
patch-clamp configuration before and after optogenetic and electrical stimulation (*Figure 5*).

We first measured membrane potential dependency to current injections with I-V plots in pyramidal
neurons with opsin expression. To do this, we kept the membrane potential at –70 mV and applied
series of 500ms current injections ranging from –100 pA to 160 pA, with 20 pA alternating positive and
negative steps (black traces; *Figure 5A*). We first observed a significant increase in evoked number
of spikes after optogenetic stimulation (blue traces; 5±1–6±1 spikes; p=0.015 by Wilcoxon; n=12;
*Figure 5D–a*). These increases in firing rate were not observed in unstimulated neurons (5±1 and 5±1;
p=1 by Wilcoxon; n=7; *Figure 5D–b*). We also noticed that neurons expressing opsin exhibited a
small alteration in their intrinsic properties. In particular, the firing rate became higher than in neurons

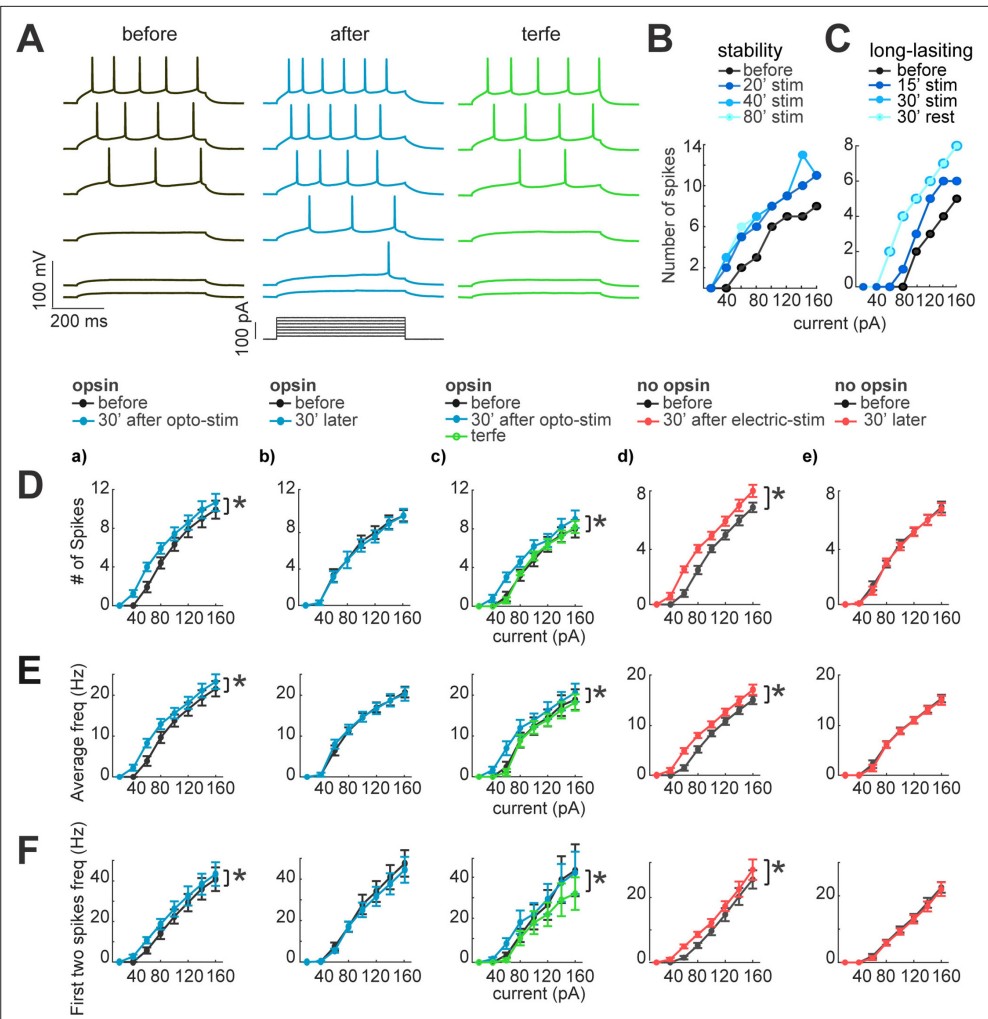

**Figure 5.** Neuronal excitability increases after optogenetic or electrical stimulation. (**A**) Current dependence firing rate increases after optogenetic or electrical stimulation. Firing rate increases returns to basal conditions after the application of terfenadine. Representative membrane voltage recordings with perforated patch-clamp (top) in response to 500ms series of 20 pA current steps (20–120 pA, bottom). Black traces show membrane potential in response to current steps before stimulation. Blue traces represent membrane voltage recordings of neurons with opsin expression and responses after 30 min of optogenetic stimulation. The green traces show effect of the terfenadine (10 µM) in membrane potential post-stimulation. (**B**) Decrease in rheobase after stimulation is stable. Current-dependent firing rate increased in a representative neuron after 20 min of continuous optogenetic stimulation. This firing rate did not show more changes even after 80 min of continued stimulation (p=0.4 by Wilcoxon; n=7 neurons, 5 mice). (**C**) Decrease in rheobase after stimulation is long-lasting. Gradual increase in firing rate of a representative neuron after 15 min of optogenetic stimulation and then after 30 min. Increases in firing rate were stable 30 min post-stimulation (rest), (p=0.4 by Wilcoxon; n=7 neurons, 5 mice). (**D**) Neuronal activity increase after optogenetic or electrical stimulation. For every graph (**a–e**), thicker black lines are average spikes of all neurons before stimulation protocol. Thicker lines correspond to (a) blue, optogenetic stimulation; (b) blue, 30 min without stimulation in neurons with opsin expression; (c) green, terfenadine (10 µM) effect after optogenetic stimulation; (d) red, electrical stimulation in neurons without opsin expression; (e) red, 30 min without stimulation in neurons without opsin expression. Data represent mean ± SEM. *p<0.05 by Wilcoxon. (**E**) Same as D but for average frequency for each current step. *p<0.05 by Wilcoxon. (**F**) The frequency of the first two action potentials increased for the lower current steps after stimulation. Same as D but for the average of the first two action potentials frequency. *p<0.05 by Wilcoxon.

The online version of this article includes the following source data for figure 5:

**Source data 1.** Neuronal excitability increases after optogenetic or electrical stimulation.

without opsin in control condition (5±1 firing rate of neurons with opsin expression, 3.5±1 neurons without opsin, but this difference was not significant p=0.1 by Mann-Whitney; n=7).

Neurons needed at least 15–20 min of optogenetic or electrical stimulation to increase their excitability, but, after their firing rate increased, it was stable for the 80 min duration of the experiment (6±1–6±1 spikes; p=0.5 by Wilcoxon; n=7; *Figure 5B*). The long-lasting increase in firing rate did not require continued stimulation, since after pausing optogenetic stimulation for 30 min, it remained elevated (6±1–6±1 spikes; p=0.8 by Wilcoxon; n=7; *Figure 5C*). Similar results were found for the average instantaneous firing frequency, which increased after optogenetic stimulation from 15±3–17±3 Hz; p=0.015 by Wilcoxon; n=12; *Figure 5E–a*. Unstimulated neurons did not show significant differences (from 16±3–16±3 Hz; p=0.5 by Wilcoxon; n=7; *Figure 5E–b*). Finally, to evaluate the modification of firing properties that could facilitate burst induction, we measured the mean frequency of the two first spikes for each current step, finding a significant increase (before: 24±3 Hz; after optogenetic stimulation: 27±3 Hz; p=0.015 Wilcoxon; *Figure 5F–a*), whereas unstimulated neurons did not show significant differences after 30 minutes of recordings (25±6 and 24±6 Hz; p=0.2 by Wilcoxon; n=7; *Figure 5F–b*). To rule out that these changes in firing properties were due to the illumination, we recorded neurons without opsin expression in perforated patch-clamp, before and after optogenetic stimulation. We did not detect any significant change in current injection firing-dependence (4±1 and 4±1 spikes; p=1, by Wilcoxon; n=7), nor in the average instantaneous frequency (7±2 Hz before; and 7±2 Hz after; p=0.6 by Wilcoxon). These data, similar to those from unstimulated neurons, demonstrate that the LED illumination does not induce changes in cellular excitability.

## Terfenadine reverts optogenetic stimulation increases neuronal excitability

These results revealed that optogenetic stimulation produced a robust increase in current-dependent firing. To pharmacologically explore potential mechanisms underlying these changes, we used terfenadine, a pharmacological agent that blocks currents mediated by Ether-a-go-go Related Gene (ERG) channels, and, in neocortical pyramidal cells, prevents persistent firing and increases in membrane resistance (*Cui and Strowbridge, 2018*). Indeed, the increases in firing rate and frequencies induced by optogenetic stimulation reverted to control conditions after application of 10 μM terfenadine (5±1 and 4±1 spikes; p=0.015 by Wilcoxon; n=7; *Figure 5D–c*). No differences were found comparing with before stimulation and when terfenadine was applied (4±1 and 4±1 spikes; p=0.06 by Wilcoxon; n=7). In unstimulated neurons without opsin, terfenadine generated a reduction in firing rate, consistent with a baseline activation of terfenadine-sensitive conductances (4±1 before; and 3±1 terfenadine p=0.03 by Wilcoxon; n=7). We concluded the increase in excitability due to optogenetic stimulation can be reverted by terfenadine. This could be due to blockage of baseline conductances, such as ERG channels or other targets like histamine1, kir 6, and Kv11.1 (*Carter et al., 1985*).

## Electrical stimulation increases neuronal excitability

After characterizing the changes in firing properties due to optogenetic activation, we explored if single-cell electrical stimulation also generated similar alterations. Electrical stimulation via a patch pipette avoids possible side effects produced by the optogenetic stimulation, or by the activation of an undetermined number of opsin-expressing neurons. Electrical stimulation of one neuron at the time also rules out synaptic effects. In these single-cell electrical stimulation experiments, we observed similar effects as in optogenetic stimulation: increases in firing rate (before 4±1 spikes; after 5±1 spikes; p=0.015 by Wilcoxon; n=22; *Figure 5D–d*); increases in average frequency (8±2 Hz before; 10±2 Hz after; p=0.015 by Wilcoxon; *Figure 5E–d*); and increases in frequency of the first two spikes (14±3 Hz before; 17±3 Hz after; p=0.015 by Wilcoxon; *Figure 5F–d*). Unstimulated neurons did not show any changes in firing rate (before 4±1; after 4±1 spikes; p=1 by Wilcoxon; n=10; *Figure 5D–e*); or average frequency (9±2 Hz before; and 9±2 Hz after; p=1 by Wilcoxon; *Figure 5E–e*); nor in the first frequency (15±3 Hz before; and 15±3 Hz after; p=1 by Wilcoxon; *Figure 5F–e*).

The consistency in results from optogenetic and electrical stimulation indicate that the increases in cellular excitability are not due to opsin expression or optogenetic protocol, and is a property of individual neurons that can be induced by stimulation of individual cells.

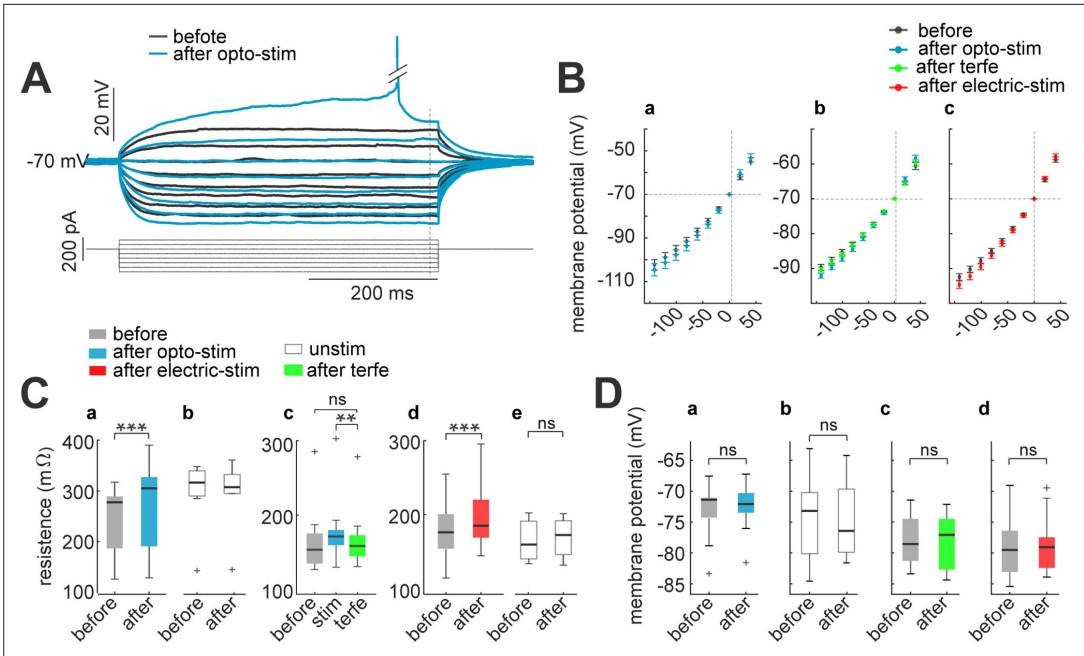

**Figure 6.** Membrane input resistance increases after optogenetic or electrical stimulation. (**A**) Membrane resistance increases after optogenetic or electrical stimulation; these returns to control condition after terfenadine application. Representative membrane voltage recordings with perforated patch-clamp in response 500ms series of 20 pA current pulses from 40 to –100 pA (bottom). Black traces show membrane potential before optogenetic stimulation, and blue traces correspond to 30 min after (**B**) I-V plots show relation between current injection and membrane potential. For every graph (a–c), black dots are average voltage membrane response to current steps before stimulation protocol. Blue average voltage membrane corresponds to (a) optogenetic stimulation; (**b**) green, effect of terfenadine (10 µM) after optogenetic stimulation; (**c**) red, electrical stimulation in neurons without opsin expression. Data represent mean ± SEM. (**C**) Membrane resistance increased after optogenetic or electrical stimulation. (**a**) Optogenetic stimulated neurons: median and blue box plot; p=0.0001 by Wilcoxon; n=10 neurons, 9 mice. (**b**) Unstimulated neurons with opsin expression: median and gray box plots; p=0.7 by Wilcoxon, n=13 neurons, 5 mice. (**c**) Terfenadine (10 µM) effect after optogenetic stimulation: median and green box plot; p=0.003 by Wilcoxon; n=8 neurons, 5 mice. (**d**) Electrical stimulated neurons: median and red box plot; p=0.00004 by Wilcoxon; n=22 neurons, 15 mice. (**e**) Unstimulated neurons without opsin expression: median and gray box plot; p=0.5 by Wilcoxon; n=11 neurons, 8 mice. (**D**) Resting membrane potential remain without changes after stimulation. (**a**) Optogenetic stimulated neurons: median and blue box plot; p=0.4 by Wilcoxon; n=10 neurons. (**b**) Unstimulated neurons: median and gray box plots; p=0.8 by Wilcoxon; n=13 neurons. (**c**) Terfenadine (10 µM) effect after stimulation: median and green box plot; p=0.8 by Wilcoxon; n=9 neurons. (**d**) Electrical stimulated neurons: median and red box plot; p=0.9 by Wilcoxon; n=13.

The online version of this article includes the following source data for figure 6:

**Source data 1.** Membrane voltage responses to current injections, before and after optogenetic or electrical stimulation.

**Source data 2.** Input resistance before and after optogenetic or electrical stimulation.

**Source data 3.** Resting membrane potential before and after optogenetic or electrical stimulation.

## Optogenetic and electrical stimulation increase membrane resistance

An increase in excitability could be due to an increase in membrane resistance (***Marder and Goaillard, 2006***). To calculate membrane resistance, we measured membrane potential changes produced by hyperpolarizing current pulses of low intensity (from 40 to –100 pA; 500ms duration, applied every second), before and after optogenetic stimulation (***Figure 6A***). Membrane input resistance was then measured as the slope of the voltage-current (I-V) relationship (***Figure 6B***). We observed a significant increase in input resistance from 278±18 mΩ to 305±22 mΩ after optogenetic stimulation (p=0.0001 by Wilcoxon; n=14; ***Figure 6C–a***). Meanwhile, resistances of unstimulated neurons, with opsin expression, did not show any change before (318±23 mΩ) or 30 min after stimulation (309±24 mΩ; p=0.9

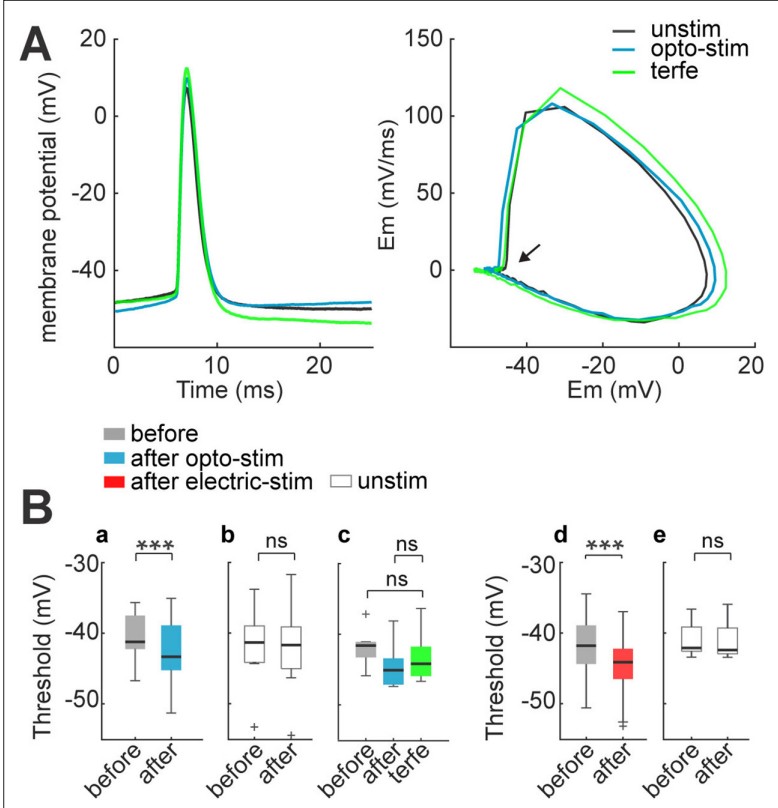

**Figure 7.** Spike threshold decreases after optogenetic and electrical stimulation. (**A**) Firing threshold shifts to more negative potentials after optogenetic and electrical stimulation. Left: firing threshold was measured as the first action potential generated with 60 pA current step: before (black), after 30 min of optogenetic stimulation (blue) and terfenadine post-stimulation (green). Right: phase plot of action potentials: before (black), after (blue), and terfenadine (green). The arrow shows firing threshold. (**B**) Firing threshold was measured for every cell before and after: (a) optogenetic stimulation, median and blue box plot (p=0.0009 by Wilcoxon, n=13 cells, 9 mice); (**b**) Unstimulated neurons with opsin expression, median and gray box plots (p=0.7 by Wilcoxon; n=8 cells, 5 mice). (**c**) Terfenadine (10 µM) post-stimulation, median and green box plot; p=0.003 by Wilcoxon; n=8 neurons, 5 mice; firing threshold before stimulation and after terfenadine (p=0.1 by Wilcoxon); (**d**) electrical stimulation, median and red box plot (p=0.00004 by Wilcoxon; n=22 cells, 15 mice). (**e**) Unstimulated neurons without opsin, median and gray box plot (p=0.7 by Wilcoxon; n=11 cells, 8 mice).

The online version of this article includes the following source data for figure 7:

**Source data 1.** Firing threshold before and after optogenetic or electrical stimulation.

by Wilcoxon; n=8; *Figure 6C–b*). These increases were reverted by the addition of terfenadine (from 172±15 mΩ after stimulation to 158±16 mΩ after terfenadine; p=0.02 by Wilcoxon; n=9; *Figure 6C–c*). Also, addition of terfenadine brought back the resistance to its pre-stimulation values, 158±15 mΩ before stimulation to 153±16 mΩ with terfenadine; p=0.9 by Wilcoxon. Furthermore, electrical stimulation in non-expressing neurons also induced a significant increase in input resistance (from 176±9– 185±9 mΩ; p=0.0003 by Wilcoxon; n=11; *Figure 6C–d*). Unstimulated neurons did not display changes in membrane resistance (before 171±11 to after 174±8 mΩ; p=0.3 by Wilcoxon; n=11; *Figure 6C–e*). Control experiment with non-expressing neurons did not show any alteration in membrane resistance after optogenetic stimulation of the slices (before 176±20 mΩ; after 179±20 mΩ; p=0.3 by Wilcoxon).

It has been reported that membrane resting potential can also change through increases in excitability, resulting in a depolarization (*Debanne et al., 2019*; *Mellor et al., 2002*; *Ross and Soltesz, 2001*). To explore this, we calculated membrane resting potential as the median of spontaneous membrane activity without any current injection. We observed that resting potential remained stable under all condition: before and after optogenetic stimulation (–73±2 to -72±1 mV; p=0.4 by Wilcoxon; n=10; *Figure 6D–a*); unstimulated neurons (–73±2 to -75±2 mV; p=0.8 by Wilcoxon; n=13;

*Figure 6D–b*). The application of terfenadine after stimulation also did not reveal significant changes in membrane potential (–78±2 to -77±2 mV; p=0.8 by Wilcoxon; n=8; *Figure 6D–c*) and no changes before and after electrical stimulation (–80±1 to -79±1 mV; p=0.9 by Wilcoxon; n=13; *Figure 6D–d*). This lack of resting membrane potential changes during intrinsic neuronal excitability agrees with previous reports (*Cudmore and Turrigiano, 2004*; *Disterhoft et al., 1986*; *Pignatelli et al., 2019*). We conclude that the increase in excitability induced by optogenetic or electrical stimulation is associated with increases in membrane resistance, but not with membrane resting potential.

## Lowering of action potential threshold after optogenetic and electrical stimulation

The changes in frequencies of the first two spikes that we observed could also be due to spike threshold changes. Burst firing depends on the interplay between the afterhyperpolarization (AHPs) and afterdepolarizations (ADPs) that follow the first action potential (*Brumberg et al., 2000*; *Mahon and Charpier, 2012*; *Paz et al., 2009*). To explore this, we evaluated the dynamics of action potentials with phase plot analysis. We used the first spikes from 60 pA current injections in the I-V dataset; because during this current step, most neurons consistently displayed action potentials before and after optogenetic or electrical stimulation protocol. Action potentials were detected after setting a threshold of >0.00015 in the first derivative of the membrane potential waveform, a value that corresponded to the firing threshold, and voltage/time values (mV/ms) were plotted as a function of membrane potential (mV). In this analysis, action potentials are represented as a loop in which the starting point represents the firing threshold (*Figure 7A*). After optogenetic stimulation, the majority of phase plots displayed a leftward shift, making evident a lowering in firing threshold (–41±1 before to –43.7±1 mV after; p=0.0012 by Wilcoxon; n=13; *Figure 7B–a*). This change was partially reverted by terfenadine (10 µM; –45±1 mV after stimulation to –44±1 mV after terfenadine; p=0.05 by Wilcoxon; n=8; *Figure 7B–c*). We found no statistically significant differences in firing threshold of unstimulated neurons, when comparing the beginning and the end of the recording (–41±2 and –41±2 mV respectively; p=0.7 by Wilcoxon; n=8 cells; *Figure 7B–b*). Also, non-expressing neurons did not show any alteration of firing threshold after optogenetic stimulation of the slices (before –40±1.4 mV to –40±1.5 mV after; p=0.7 by Wilcoxon). Thus, the LED light did not induce damage or alterations in this cell-intrinsic parameter.

We also documented a similar lowering of the threshold voltage in electrically stimulated neurons (before –42±0.8 mV; after –44±0.8 mV; p=0.00004 by Wilcoxon; n=22 cells; *Figure 7B–d*). Unstimulated neurons displayed no changes in threshold voltage (–42±0.7 mV at the first min and –43±0.8 mV 30 min later; p=0.5 by Wilcoxon; n=11; *Figure 7B–e*).

Combined increase in membrane resistance and reduction of firing threshold, can explain the increase in neuronal excitability previously observed. Thus, optogenetic and electrical stimulation significantly increases action potential firing in response to synaptic inputs, facilitating the formation of a neuronal ensemble.

## Discussion

Since Hebb, the traditional view of how a neuronal ensemble (i.e., an assembly) is created by strengthening its synaptic connections after repeated coactivation (*Bliss and Gardner-Medwin, 1973*; *Carrillo-Reid et al., 2016*; *Cossell et al., 2015*; *Morris, 1999*; *Hoshiba et al., 2017*). However, several reports have questioned whether synaptic plasticity is the only mechanism for circuit plasticity, proposing instead cell-intrinsic mechanisms (*Angelo et al., 2012*; *Brown et al., 2019*; *Disterhoft and Oh, 2006*; *Ganguly et al., 2000*; *Marder and Goaillard, 2006*; *Ryan et al., 2015*; *Zhang and Linden, 2003*). Consistent with this, an in contradiction with our own predictions that synaptic plasticity mediated neuronal ensembles (*Carrillo-Reid et al., 2016*), we describe here that simultaneous optogenetic or electrical stimulation of neurons in vitro increases the excitability of the stimulated cells. The increases in correlation among neurons, generated by an increased excitability, could also contribute to generate or enhance long-term synaptic plasticity (*Lisman et al., 2018*; *Titley et al., 2017*). Together, cellular and synaptic mechanisms could contribute to neuronal ensemble development (*Penn and Shatz, 1999*; *Zhang and Poo, 2001*), flexibility, plasticity (*Clopath et al., 2017*; *Kirkwood et al.,*

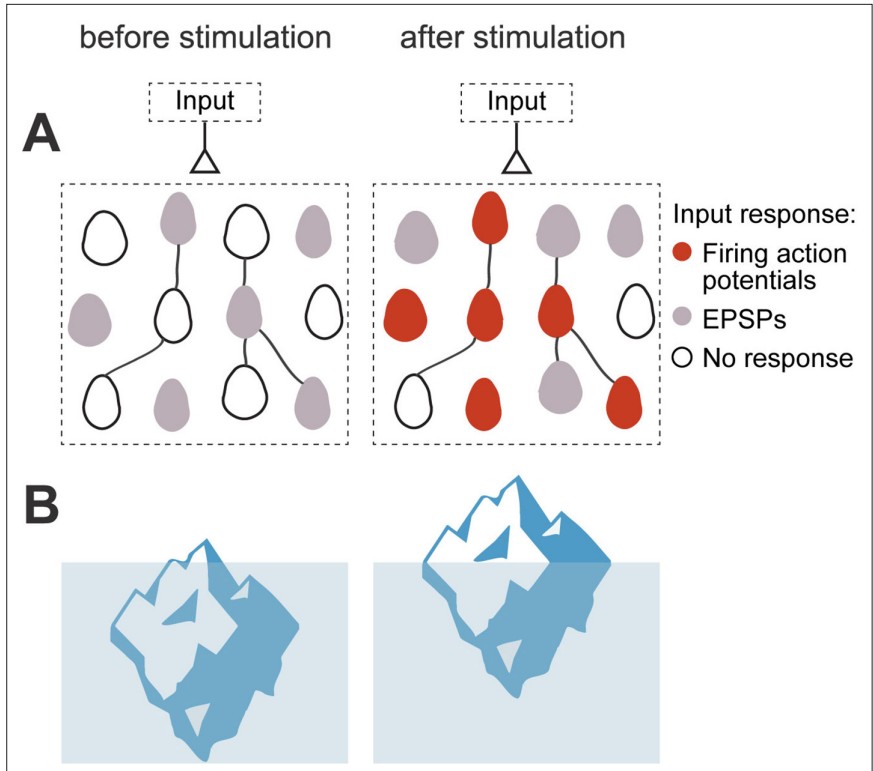

**Figure 8.** Iceberg model of ensemble formation. (**A**) Emergence of ensembles after increases in neuronal excitability. Neurons shift to a more excitable state after stimulation, so neuronal responses are amplified and the circuit now responds to an external input by activating a neuronal ensemble. Color corresponds to membrane potential response to a synaptic input: white is resting membrane potentials, gray are subthreshold responses, and red are suprathreshold ones, with firing of action potentials. Stimulated neurons become more excitable, so the same inputs induce some of them to fire (red cells), while other cells have increased subthreshold responses (gray cells). The model explains how an ensemble is formed but does not assume any changes in numbers of strength of local synapses. (**B**) Left: Iceberg emergence: An iceberg keeps underwater. Right: But if its weight decreases, the iceberg emerges above water. Weight (or density) is an intrinsic property of the iceberg and, by changing it, the iceberg changes its response to the same environment. Likewise, for a neuronal ensemble, membrane resistance and firing threshold are intrinsic neuronal properties that can be modified, and they enhance its response to the same excitatory input intensity, resulting in an increased depolarization and generation of action potentials.

*1996*; *Mrsic-Flogel et al., 2007*; *Pérez-Ortega et al., 2021*; *Wandell and Smirnakis, 2009*; *Wiesel and Hubel, 1963*).

## Role of synaptic mechanisms in ensemble formation

Our experimental procedures, similar to those that build ('imprint') neuronal ensembles in vivo (*Carrillo-Reid et al., 2016*), lead to increases in correlations of spontaneous activity, a hallmark of a neuronal ensemble, just as it does in vivo. Using paired recordings, we find similar rates of connectivity between neuronal pairs as previous reports (*Cossell et al., 2015*; *Holmgren et al., 2003*; *Ko et al., 2011*; *Lefort et al., 2009*; *Levy and Reyes, 2012*; *Markram et al., 1997*; *Song et al., 2005*). Also, consistent with many reports, connected neurons display higher correlation in spontaneous activity than unconnected ones (*Cossell et al., 2015*; *Ko et al., 2011*; *Yoshimura et al., 2005*). Interestingly, our results also showed that 50% of unconnected neurons can be as correlated as connected neurons, and contrary to our own expectations, we did not detect any new connections formed after optogenetic stimulation in previously unconnected neurons.

Based on previous studies, there is preferential connectivity between neurons with similar receptive fields (*Cossell et al., 2015*; *Ko et al., 2011*; *Yoshimura et al., 2005*). Increased synaptic connectivity also probably underlies pattern completion in cortical ensembles (*Carrillo-Reid et al., 2019*; *Pignatelli et al., 2019*), and a few strong connections could drive local excitation of the majority of

neurons with weak connectivity (*Cossell et al., 2015*; *Holmgren et al., 2003*; *Ko et al., 2011*; *Lefort et al., 2009*; *Levy and Reyes, 2012*; *Markram et al., 1997*; *Song et al., 2005*; *Figure 8*). Consistent with these hypotheses, we expected to observe synaptic efficiency changes between connected neurons after co-stimulation (*Bliss and Gardner-Medwin, 1973*). But, rather than synaptic potentiation, our stimulation protocols induced a moderate biphasic plasticity of connected neurons and also in the local circuit, with an initial depression followed by a recovery potentiation of evoked EPSCs and EPSPs after 20 min of rest.

This result agrees with reports that EPSP amplitude can decrease after high stimulation frequencies, but recovers from depression by using lower stimulation rates and, after a period of rest can rebound which is revealed as a potentiation (*Feldmeyer et al., 2006*). We cannot rule out that Hebbian synaptic plasticity contributes to this potentiation. However, an alternative explanation is that this could instead be due to a depletion, followed by a recovery, of the ready releasable pool of synaptic vesicles in the presynaptic neuron (*Feldmeyer et al., 2006*; *Schneggenburger et al., 2002*).

In addition to these synaptic changes, our results demonstrate that persistent and simultaneous stimulation robustly increases membrane resistance. Thus, neuronal excitability could increase the efficiency of existing synaptic connections between neurons in an ensemble, without significant synaptic plasticity. Indeed, we find that spontaneous EPSPs become amplified in single optogenetic and electrically stimulated neurons. A larger synaptic input could also naturally result from an overall increase in neuronal firing too, since increased neuronal excitability would result in increase in neurotransmitter release or amplified EPSPs. This agrees with reports that reveal that synaptic inputs can be enhanced by intrinsic membrane mechanisms (*Campanac and Debanne, 2008*; *Hu et al., 2002*; *Li et al., 2020*; *Manabe et al., 1992*; *Marder and Goaillard, 2006*; *Narayanan and Johnston, 2008*; *Turrigiano et al., 1998*). For all these reasons, we suspect the modest synaptic plasticity that we observe can be explained by the effects of the stimulation protocols in the ready releasable pool and intrinsic changes in excitability in the stimulated neurons and the network.

Primary visual cortex (V1) responds to sensory information. Thus, one of the main roles of V1 neurons is to integrate sensory inputs (*Buonomano and Merzenich, 1998*; *Mountcastle, 1997*; *Figure 8*). Unitary EPSP, that is, pyramidal to pyramidal monosynaptic connections, would not be expected to increase the firing probability of postsynaptic neurons, in vitro or in vivo (*Holmgren et al., 2003*; *Jouhanneau et al., 2018*). However, the joint activation of several neurons in the local circuit (in our case: neighbor neurons with opsin expression), could evoke compound EPSC and EPSP with amplitudes larger than unitary EPSC or EPSP, and could generate synfire chains of activity that propagate through the cortex (*Abeles, 1991*). Thus, increases in synaptic efficacy, mediated by an increase in neuronal excitability of connected neurons, could play a significant role in the local circuit, improving the efficiency of the flow of the information during visual stimulation and during the recall of neurons associated with a memory.

## Role of intrinsic mechanisms in ensemble formation

Changes in intrinsic properties are known to alter EPSP amplitude, for example, by the modification of membrane conductance induced by postsynaptic stimulation (*Hu et al., 2002*; *Li et al., 2020*; *Manabe et al., 1992*; *Marder and Goaillard, 2006*; *Narayanan and Johnston, 2008*; *Turrigiano et al., 1998*). These changes might be due to modification in intrinsic membrane conductances (*Li et al., 2020*; *Manabe et al., 1992*; *Narayanan and Johnston, 2008*), and they could amplify synaptic inputs and improve the accuracy of synaptic integration of neurons (*Desai et al., 1999*; *Mahon and Charpier, 2012*; *Malik and Chattarji, 2012*; *Sjöström et al., 2008*; *Turrigiano et al., 1994*).

Confirming the importance of intrinsic mechanisms, in our experiments, robust increases in firing rate were easily detected in optogenetic and electrically stimulated neurons. Even though we noticed that the expression of opsin by itself can have small effects in excitability, similar changes were observed in non-expressing neurons after electrical stimulation. This rules out that the changes in excitability observed were due to the opsin expression. An increase in cellular activity through cell-intrinsic mechanisms after prolonged firing agrees with many functional studies in vertebrates, using different protocols (*Abraham et al., 2019*; *Ryan et al., 2015*; *Titley et al., 2017*). In fact, increases in excitability are also found following LTP induction in visual cortical neurons (*Cudmore and Turrigiano, 2004*), in hippocampal CA1 pyramidal cells (*Campanac and Debanne, 2008*; *Ganguly et al., 2000*; *Xu et al., 2005*), and after behavior training in CA1 (*Disterhoft et al., 1986*). Intrinsic excitability is

also modified when an animal is exposed to novel sensorial experience (*Brown et al., 2019*) or after environmental enrichment (*Valero-Aracama et al., 2015*).

The intensity-dependent and persistent increase in firing rate after stimulation protocol could be due to changes in membrane resistance and firing threshold. The synergy between these mechanisms could make neurons reach firing threshold with smaller depolarizations. Thus, an increase in excitability would be particularly relevant for lower current injections generated by weaker inputs. We propose that, after optogenetic or electrical stimulation, neurons shift to a more excitable state, like an iceberg emerging out of the water (*Figure 8*). Therefore, the same unchanged synaptic inputs could bring neurons to threshold and induce increased output firing after repetitive stimulation, improving the synaptic efficiency in pyramidal neurons (*Debanne et al., 2019*; *Lisman et al., 2018*; *Nicoll et al., 1993*; *Titley et al., 2017*; *Zhang and Linden, 2003*). Thus, neuronal optogenetic and electrical stimulation might alter the circuit without changing the synapses themselves, and this phenomenon could occur quickly.

## Mechanisms of increased excitability in ensemble formation

Modifications in sub- and suprathreshold membrane conductances that initiate an action potential could underlie increased excitability (*Brumberg et al., 2000*). In terms of passive properties, our results coincide with many studies that have shown alterations in input resistance as the mechanism responsible for increases in excitability (*Aizenman and Linden, 2000*; *Aou et al., 1992*; *Armano et al., 2000*; *Brown et al., 2019*; *Disterhoft et al., 1986*; *Marder and Goaillard, 2006*; *Woody et al., 1991*; *Xu et al., 2005*). Activity-dependent increases in input resistance have been shown in hippocampal neurons. There, the increase in excitability could be generated by suppression of G-protein-coupled inwardly rectifying K⁺ channels (GIRK channels) (*Valero-Aracama et al., 2015*); or by cAMP-responsive element-binding protein (CREB)-dependent control of excitability by reducing K⁺ conductance. (*Lisman et al., 2018*; *Yu et al., 2017*) or potassium and sodium voltage gate conductances (*Campanac and Debanne, 2008*; *Ganguly et al., 2000*). Additionally, the contribution of H-channels in intrinsic membrane properties (*Campanac et al., 2008*; *Fan et al., 2005*). However, that voltage sag, which corresponds to current-H, is largely absent in layer 2/3 pyramidal neurons (*Kalmbach et al., 2018*; *van Aerde and Feldmeyer, 2015*). Our experiment showed only a few neurons that displayed a modest current-H, but it was not affected by optogenetic or electrical stimulation (not shown).

Moreover, recent studies show that persisting firing could be mediated by an Ether-à-gogo Related Gene (ERG) K⁺ channel in neocortical pyramidal neurons (*Cui and Strowbridge, 2018*; *Debanne et al., 2019*). In this view, calcium entry induced by repetitive neuronal action potential could modulate leak potassium currents in the neurons, which will generate persistent activity. Consistent with this, in our experiments, the blockage of ERG channels by terfenadine returned the firing rate and the increased input resistance induced by optogenetic and electrical stimulation to basal conditions. Thus, ERG channels appear to have a tonic activity in pyramidal layer 2/3 neurons.

The cell-intrinsic mechanisms that induce excitability are cell-type and also, stimulation protocol-dependent (*Angelo et al., 2012*). Hence, different changes in action potential firing have been identified. One of the most common changes is a reduction in after-hyperpolarization (AHP) amplitude, described in hippocampal neurons (*Disterhoft et al., 1986*; *Disterhoft and Oh, 2006*; *Malik and Chattarji, 2012*; *Pignatelli et al., 2019*; *Zhang and Linden, 2003*). In our experiments, we found a significant decrease in spike threshold as the most robust change in optogenetic and electrical stimulated neurons. This parameter also determines the firing frequency of neurons associated with voltage-dependent sodium or calcium currents. Changes in threshold appear to be usual in cortical neurons after intrinsic plasticity induction (*Brumberg et al., 2000*; *Cudmore and Turrigiano, 2004*; *Mahon and Charpier, 2012*; *Paz et al., 2009*), hippocampal neurons (*Malik and Chattarji, 2012*; *Valero-Aracama et al., 2015*; *Xu et al., 2005*), and cerebellar Purkinje cells (*Aizenman and Linden, 2000*; *Armano et al., 2000*). Consistent with this, visual deprivation increases spike threshold in pyramidal neurons of visual cortex, as an example of a mechanism that reduces neuronal excitability (*Brown et al., 2019*). Also, the persistent sodium current could underlie the lowering of spike threshold, throughout the modulation of protein kinase C (*Astman et al., 1998*; *Ganguly et al., 2000*; *Valero-Aracama et al., 2015*); or protein kinase A activation-dependent on calcium influx (*Cudmore and Turrigiano, 2004*). Finally, potassium channels Kv1 can regulate the action potential threshold (*Feria*

*Pliego and Pedroarena, 2020*). Therefore, the intrinsic plasticity that we report could result from modifications of voltage-gated ion channels and inward rectification by potassium channels. Future studies will examine the exact molecular mechanisms that underlying these increases in excitability.

## Contribution of intrinsic excitability and synaptic plasticity to circuit plasticity

The apparently contradictory evidence supporting Hebbian synaptic plasticity or intrinsic mechanisms of circuit modification and ensemble formation could be reconciled if one considers the experimental protocol used, as well as temporal and mechanistic factors. As a note of caution, we should mention that the stimulation protocols we used are artificial, and the imprinting of ensembles with optogenetics or electrical stimulation may not be representative of the natural ensembles found and created in the cortex under physiological conditions. Thus, the balance of mechanisms we find may differ with in vivo ensembles, under naturalistic plasticity conditions. In particular, the optogenetic imprinting model for ensemble formation we used occurs quickly in vivo, as soon as we can measure it, within a few minutes. We also see fast changes in neuronal correlations in vitro after optogenetic and electrical stimulation. Although these fast changes are explained by fast intrinsic modifications in excitability, imprinted ensembles could be different from the circuit modification that occur during learning, which, for visual tasks, can take several days or weeks in the same circuit we study (*Carrillo-Reid et al., 2019*), and which could engage long-term plasticity, Hebbian or not Hebbian.

In addition, intrinsic plasticity can underlie or contribute to Hebbian plasticity by enhancing the probability for subsequent LTP induction and form or stabilize ensembles/engrams (*Debanne et al., 2019*; *Lisman et al., 2018*). Alternatively, intrinsic plasticity could add neurons into ensembles, even when synaptic weights do not change (*Hansel and Disterhoft, 2020*; *Titley et al., 2017*). A third possibility is that behavioral training could engage Hebbian mechanisms that would then induce increases in neuronal excitability (*Disterhoft et al., 1986*; *Malik and Chattarji, 2012*; *Valero-Aracama et al., 2015*). So, circuit plasticity could be due to the interplay of synaptic and intrinsic mechanisms and this balance could be different for short and long-term plasticity. In this view, ensembles could be quickly formed by fast intrinsic mechanisms but then be modified by slower synaptic plasticity, perhaps in relation with learning.

To conclude, our work demonstrates that significant and generalized increases in cellular excitability by intrinsic mechanisms are found during ensemble formation, induced by optogenetic and electrical stimulation. The possibility of modifying input and output patterns of neurons through changes in its firing properties could increase synaptic efficacy (*Brumberg et al., 2000*; *Lisman, 1997*) and the correlation between neurons, as we have demonstrated. This could provide an ideal setting for follow-up Hebbian synaptic plasticity than could occur more slowly. As a form of cellular memory, intrinsic excitability could also implement temporary stimulus-response mappings and might play a role in rapid cognitive flexibility (*Pang and Fairhall, 2019*). Faster 'online' plasticity via intrinsic excitability could have a broad impact on network dynamics and could serve as a critical information-storage mechanism that may contribute to memory formation (*Lisman et al., 2018*; *Marder and Goaillard, 2006*; *Pérez-Ortega et al., 2021*; *Pignatelli et al., 2019*; *Xu et al., 2005*) and to the generation of intrinsic circuit states, such as ensembles and circuit attractors (*Buzsáki, 2010*; *Carrillo-Reid et al., 2019*; *Carrillo-Reid et al., 2016*; *Hopfield, 1982*; *Miller et al., 2014*).

## Materials and methods

All procedures were performed by following the U.S. National Institutes of Health and Columbia University Institutional Animal Care and Use Committee guidelines (IACUC, Protocol #AC-AAAV3464). Experiments were carried out on C57BL/6 transgenic mice (Vglut1-Cre, Jackson Laboratories; RRID:IMSR_JAX:00064) and electroporated mice CD-1 (Charles River) of both sexes at postnatal 4–8 weeks-old. Animals were housed on a 12 hr light-dark cycle with food and water ad libitum.

### Viral injection

After 1 month postnatal, animals were anesthetized with 2% isoflurane on a head-fixed stereotactic apparatus. After sterilizing the incision site, the skin was opened and, using an FG 1/4 dental drill, a small hole goes thin in the skull over the visual primary cortex (2.5 mm lateral and 0.3 mm anterior

from the lambda, 200 µm from pia). We injected 300 nL of the virus at a rate of 30–40 nL/s, using a microsyringe pump (Micro 4), and a Hamilton (7653–01) glass pipette. Once the pipette was placed, we waited 5 min before and after the virus injection and closed the scalp with sutures. Viruses injected were: AAVDJ-CaMKIIa-C1V1(E162T)-TS-p2A-EYFP-WPRE (Stanford University Gene Vector and Virus Core). Mice were used for electrophysiology experiments 2–3 weeks postinjection. These mice were used only to measure evoked EPSC of optogenetically stimulated neurons.

## In utero cortical electroporation

Pregnant mice (CD-1, Charles River) were placed in a sterile environment following *dal Maschio et al., 2012*. We performed a laparotomy, under 2% isoflurane anesthesia, and injected (0.5 mg/µL) of pCAG ChroMe-mRuby-ST (*Mardinly et al., 2018*), ~1 µL per mouse with a glass pipette, in the left lateral ventricle of embryonic day 17 pups. After that, we placed the electrodes on the head 5 mm diameter platinum disk electrodes, (Nepa Gene #CUY650P5) to electroporate with a set of three electrical pulses for pore formation (50 V; duration, 10ms; intervals, 50ms) and then 3 electrical pulses for transferring the plasmid (8 V; duration, 10ms; interval, 50ms). Finally, the embryos were returned to the abdominal cavity and left for their normal development until postnatal day 30.

## Brain slices

Mice were anesthetized with ketamine/xylazine, after transcardial perfusion procedure and posterior cervical dislocation, to obtain brain sagittal slices as described (*Ting et al., 2018*). Brains were quickly dissected and cooled in a continuously gassed (95% $O_2$ and 5% $CO_2$) icy 194 Sucrose, 30 NaCl, 2.6 KCl, 26 NaHCO$_3$, 1.2 NaH$_2$PO4, 10 glucose, 2 MgSO$_4$, and 0.2 CaCl$_2$ Titrate pH to 7.3–7.4 with HCl. 300 µm thick sagittal slices were cut on a Leica VT1200 S vibratome (Leica Biosystems) and allowed to recover for 1 hr at 34 °C aCSF solution (in mM): 195 NaCl, 2.5 KCl, 1.2 NaH$_2$PO$_4$, 26 NaHCO$_3$, 12 glucose, 1 thiourea, 1 Na-ascorbate, 1 Na-pyruvate, 2 CaCl$_2$, and 1 MgSO$_4$. Titrate pH to 7.3–7.4. Finally, slices were transferred to room temperature.

## Patch-clamp recordings

Brain slices were carefully placed in the recording chamber on an upright microscope (Olympus, BX50WI), and continuously perfused with gassed (95% O2 and 5% CO2) aCSF at 5 ml/min. Electrophysiological recordings were performed at 30 degrees, using patch pipettes of borosilicate glass (World Precision Instruments). Pipettes were pulled with a micropipette puller (DMZ-Universal puller) to a 5–6 MOhm as final resistances.

For whole-cell and perforated patch-clamp, patch pipettes were filled with intracellular solution containing the following (in mM): 130 K-gluconate, 20 KCl, 10 HEPES, 10 Na$_2$-phosphocreatine, 4 Mg-ATP, 0.03 Na$_2$-GTP, and titrated to pH 7.3 with KOH. Gramicidin (sigma Aldrich) was dissolved in dimethyl sulfoxide (1 mg /10 µl DMSO) and then dissolved with intracellular solution ~20 ng/ml. To allow membrane sealing, the tip of the pipette was first filled with clean solution and then the pipette was back-filled with a gramicidin-containing solution.

In each case, the patch pipettes were placed in contact with the cell with a MPC-200 micromanipulator (Sutter Instrument). Stimulation and data acquisition were sampled at 10 kHz and low pass filtered at 4 kHz using a Multiclamp 700B amplifier (Molecular Devices) and Im-Patch open-access software http://impatch.ifc.unam.mx. Recordings were discontinued when leaking current was >25 pA or seal resistance <1 GΩ. Recordings were analyzed with custom routines in MATLAB.

## Imaging

To identify target brain region, we used an upright microscope with a 4 X/0.10 NA air objective (Olympus) before switching to 60 X/0.90 W NA water immersion objective to confirm ChroME or C1V1 opsin expression on target cells. For ChroME, the mRuby3 fluorescent reporter was excited with monochromatic light transmitted through a fiber optic into the microscope (Olympus 100 w high-pressure mercury burner model BH2RFLT3). Emitted fluorescence was band-passed with an Olympus U-49006 ET CY5 filter set: 620/60 excitation filter, 520 dichroic mirrors, and 700/75 emission filter. For C1V1, the EYFP fluorescent reporter was visualized via a 500/24 ex, 520 dichroic mirrors, and a 542/27 em (Semrock). Fluorescence images were acquired (50ms exposure; 10 Hz) using a camera (Orca-ER C4742-95, Hamamatsu) and shutter (UNIBLITZ model VCM-D1) controlled by HCImage software

(Hamamatsu). We visualized individual neurons' fluorescence deep (100 µm) into the brain slice in layer 2/3, evidence of C1V1 or mRuby3 ChroMe expression.

## Optogenetic stimulation

Optogenetic stimulation of ChroMe opsin was performed with a 470 nm fiber-coupled LED (M470F1, Thorlabs), fiber optic cannula (M79L01, Thorlabs), and LED driver (M00329012, Thorlabs). Optogenetic stimulation of C1V1 was performed using a 617 nm fiber-coupled LED (M617F2, Thorlabs), fiber optic cannula (CFM14L10, Thorlabs), and LED driver (DC2200, Thorlabs). The optogenetic stimulation protocol consisted of trains of 10 Hz, 5ms light pulses for 4 s followed by a 10 s rest, we repeated the same stimulus during 15–80 min to mimic stimulation conditions previously used in vivo experiments (*Carrillo-Reid et al., 2016*) to co-stimulate neurons.

Only neurons that exhibited action potentials in response to each LED pulse were included in the data analysis of optogenetically stimulated neurons. LED intensity applied throughout the protocol was the lowest necessary for inducing action potentials. For optogenetically stimulated neurons, parameters were monitored before and 15–80 min after the stimulation protocol. Optogenetic stimulation protocol was stopped every 15 or 20 min to monitor parameters, and then restarted. For unstimulated neurons, the same parameters were monitored at the beginning and then every 15–20 min for 80 min approximately.

We observed that neurons expressing opsin often exhibited small alterations in intrinsic properties. Specifically, the firing rate was higher in neurons with opsins than in control cells, and the membrane resistance and membrane potential become higher and more variable than neurons without opsin expression.

## Experimental design

To co-activate neurons (*Carrillo-Reid et al., 2016*), we delivered C1V1 or ChroME opsin in pyramidal neurons (*Mardinly et al., 2018*). No significant differences were found in experiments with both opsins and data were pooled together. We localized layer 2/3 pyramidal neurons in primary visual cortex of brain slices expressing opsin by virus injection and in utero electroporation (*Figure 1A*). To test whether the co-stimulation of connected neurons reinforced or created new connections (*Carrillo-Reid et al., 2016*), we examined pairs of connected and unconnected neurons (*Figure 1B–C*). We recorded and characterized evoked excitatory postsynaptic currents and potentials (EPSCs and EPSPs). We also recorded passive and active electrical properties of neurons using perforated patch-clamp recording. After measuring these parameters in control condition, we began the optogenetic or electrical stimulation protocol (*Figure 1D and E*). For optogenetic stimulation, action potentials were generated by trains of 10 Hz, 5ms light pulses for 4 s followed by a 10 s rest (*Figure 1D*). For electrical stimulation protocol, action potentials were generated with 10 Hz trains, 5–10ms depolarizing currents pulses 400–600 pA for 4 s followed by a 10 s rest (*Figure 1E*). During these experiments, we noticed that the time after optogenetic or electrical stimulation was important, immediately after stimulation, the membrane activity of stimulated neurons became unstable, particularly during the first 3 min after stimulation. After that, membrane potential recovered. Thus, we evaluated membrane parameters 3–5 min post-stimulation and synaptic parameters 10 min post-stimulation.

## Statistical analysis

Statistical details are showing in each figure legends. Group data are expressed as median ± sem. Wilcoxon and Mann-Whitney test were used for nonparametric analysis. Differences between two groups were considered significant when $*p<0.05$, $**p<0.01$ and $***p<0.001$. All statistics were performed using statistical functions in MATLAB.

## Acknowledgements

We thank James Holland, for his assistance and members of the Yuste Lab for useful comments. Supported by R01EY011787 and R01MH115900. TA and JP have postdoctoral fellowships from the National Council of Science and Technology from Mexico (CONACYT). SK and RY conceived the project. SK and TA performed experiments and TA and RY wrote the paper. TA, SK and JP analyzed the data. All authors planned experiments, discussed results and edited the paper. RY assembled and directed the team and secured funding and resources.

## Additional information

### Funding

| Funder | Grant reference number | Author |
|---|---|---|
| National Institute of Mental Health | R01EY011787 | Rafael Yuste |
| National Institute of Mental Health | R01MH115900 | Rafael Yuste |
| Consejo Nacional de Ciencia y Tecnología | 287725 | Tzitzitlini Alejandre-García |
| Consejo Nacional de Ciencia y Tecnología | CVU365863 | Jesús Pérez-Ortega |

The funders had no role in study design, data collection and interpretation, or the decision to submit the work for publication.

### Author contributions

Tzitzitlini Alejandre-García, Conceptualization, Data curation, Formal analysis, Investigation, Methodology, Writing – original draft, Writing – review and editing; Samuel Kim, Data curation, Formal analysis, Investigation, Methodology; Jesús Pérez-Ortega, Conceptualization, Investigation, Software, Visualization; Rafael Yuste, Conceptualization, Funding acquisition, Investigation, Project administration, Supervision, Validation, Visualization, Writing – review and editing

### Author ORCIDs

Tzitzitlini Alejandre-García ![ORCID] http://orcid.org/0000-0002-2243-8703
Jesús Pérez-Ortega ![ORCID] http://orcid.org/0000-0001-8502-1692
Rafael Yuste ![ORCID] http://orcid.org/0000-0003-4206-497X

### Ethics

All procedures were performed by following the U.S. National Institutes of Health and Columbia University Institutional Animal Care and Use Committee guidelines (IACUC, Protocol #AC-AAAV3464).

### Decision letter and Author response

Decision letter https://doi.org/10.7554/eLife.77470.sa1
Author response https://doi.org/10.7554/eLife.77470.sa2

## Additional files

### Supplementary files

• Transparent reporting form

### Data availability

Data have been deposited with Dryad (https://doi.org/10.5061/dryad.j6q573ngc).

The following dataset was generated:

| Author(s) | Year | Dataset title | Dataset URL | Database and Identifier |
|---|---|---|---|---|
| Alejandre-García T, Kim S, Pérez-Ortega J, Yuste R | 2022 | Intrinsic excitability mechanisms of neuronal ensemble formation | https://dx.doi.org/10.5061/dryad.j6q573ngc | Dryad Digital Repository, 10.5061/dryad.j6q573ngc |

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
