## [Editor Report]

This paper provides new insights regarding how the intrinsic excitability of neurons contributes to the formation of cortical ensembles, which may underlie memory formation. Previous work had left the relative contribution of intrinsic versus synaptic changes to neural ensemble formation incompletely understood. By using an interdisciplinary approach, the authors reveal the mechanisms by which changes in intrinsic excitability may play a unique role in the formation of new neural ensembles.

---

## [Decision Letter]

**Decision letter after peer review:**

Thank you for submitting your article "Intrinsic excitability mechanisms of neuronal ensemble formation" for consideration by *eLife*. Your article has been reviewed by 2 peer reviewers, and the evaluation has been overseen by a Reviewing Editor and Ronald Calabrese as the Senior Editor. The following individual involved in the review of your submission has agreed to reveal their identity: Christian Hansel (Reviewer #3).

Essential revisions:

The reviewers agreed this paper was impactful and important. Below are the key revisions needed for publication. The reviewers also noted some new experiments that could increase the impact of the conclusions but these are not required for the revision at *eLife*.

1. It is not entirely clear why optogenetic and electrical stimulations do not lead to identical results in terms of synaptic plasticity in Figure 2 but that optogenetic stimulation in Figure 3 produces synaptic plasticity. Please clarify.

2. A few key references about intrinsic plasticity are missing in this paper. First, Ganguly et al., Nat Neurosci 2000 (and confirmed by Li et al., Neuron 2004) showed that synaptic plasticity induced by pre-and postsynaptic activity was associated with an increase in presynaptic excitability. In addition, Campanac and Debanne J Physiol 2008 showed that positive pre-post correlation induced an increase in postsynaptic firing (that is selective to the paired input) in CA1 pyramidal neurons. These papers should be quoted in this manuscript as they are directly related to the purpose of this manuscript.

3. Line 79 f: It is not clear from this paragraph, which of the cited papers provide experimental details and which one presents the 'alternative hypothesis' (Titley et al., 2017; see above). This should be described more accurately to share the precise status quo of this research field with the audience.

4. In Figure 2, the authors show that both the optical stimulation as well as the electrical stimulation trigger synaptic plasticity, consisting of an immediate depression, followed after a pause by a potentiation. This potentiation is – in the case of electrical stimulation – significantly different from the baseline values (not significant for the opto group, but the number of recordings is quite low). It thus is conceivable that this effect contributes to the enhanced correlation of activity in the network that is shown in Figure 1. While new experiments could better assess whether the enhanced correlation can persist with only excitability changes being available as a cellular mechanism, these are not necessary for publication in *eLife*. Even so, we would request that the authors discuss this potential issue in the manuscript.

*Reviewer #2 (Recommendations for the authors):*

I believe that this is a strong paper with a lot of potential for very high impact, but it all depends on whether it can be demonstrated that in the true absence of synaptic gain changes (or with those being at least reduced) the observed ensemble effect persists -- only with intrinsic plasticity as an available cellular mechanism.

This will take a bit of time and work, but it could make the paper a game-changer for plasticity research.

---

## [Author Response]

Essential revisions:The reviewers agreed this paper was impactful and important. Below are the key revisions needed for publication. The reviewers also noted some new experiments that could increase the impact of the conclusions but these are not required for the revision at eLife.1. It is not entirely clear why optogenetic and electrical stimulations do not lead to identical results in terms of synaptic plasticity in Figure 2 but that optogenetic stimulation in Figure 3 produces synaptic plasticity. Please clarify.

The results are not identical but consistent, and the difference may be due to the low n for perforated recordings from pairs of connected opsin-expressing cells. The optogenetically-stimulated group revealed trend towards depression right after the stimulation, followed with a trend towards potentiation after 20 minutes of rest. These trends occurred in 6 to 7 neurons but were non-significant. These experiments are very challenging, since they involve patch-clamp recordings from connected, opsin-expressing neurons in adult animals. Moreover, based on our electrically stimulation experiments, perforated patch is necessary, making these experiments even more challenging. In 2 of 2 connected pairs with perforated patch from opsin expressing neurons, we observe depression, recovery and potentiation in evoked EPSC after 20 min post-stimulation, similar to electrically stimulated connected pairs and similar to the optogenetic stimulation of the local circuitry. Thus, we strongly suspect that adding more optogenetically stimulated neurons would reveal identical changes in synaptic plasticity as in the electrically stimulated cells. At the same time, finding pairs of healthy connected neurons with opsin expression and record for long periods of time, reduces the probability to success this specific experimental procedure. This is different to electrically stimulated connected pairs and optogenetic stimulation of the local circuit, where the opsin expression in the recorded neuron was not necessary. That is the reason why we buttress our claim on the larger n from electrical stimulated cells, but would still like to include the optogenetically stimulated neurons, with these caveats.

We include these caveats this in the Result section:

"Although average EPSC_1_ peak currents of optogenetically stimulated neurons did not show significant changes with simulation, they revealed a trend towards amplitude depression after the stimulation, followed with a tendency towards potentiation, after a 20 minutes rest period…"

"We conclude that electrical coactivation leads to a moderate biphasic change in monosynaptic currents, with an initial depression followed by a potentiation. These results are consistent with the trends found in pairs of optogenetic stimulated neurons, where EPSC_1_ amplitude after a post-stimulation recovery period increased from 9 ± 7 pA to 19 ± 3 pA, although no statistical comparison was possible due to the low n (Figure 2C). The difficulty in obtaining perforated whole cell recordings from connected pairs of neurons in opsin expressing cells from adult brain slices precluded us from increasing the n after many attempts. These difficulties were ameliorated in the electrical stimulation experiments.”

2. A few key references about intrinsic plasticity are missing in this paper. First, Ganguly et al., Nat Neurosci 2000 (and confirmed by Li et al., Neuron 2004) showed that synaptic plasticity induced by pre-and postsynaptic activity was associated with an increase in presynaptic excitability. In addition, Campanac and Debanne J Physiol 2008 showed that positive pre-post correlation induced an increase in postsynaptic firing (that is selective to the paired input) in CA1 pyramidal neurons. These papers should be quoted in this manuscript as they are directly related to the purpose of this manuscript.

We thank the reviewer for this suggestion. We have incorporated these references in the manuscript.

3. Line 79 f: It is not clear from this paragraph, which of the cited papers provide experimental details and which one presents the 'alternative hypothesis' (Titley et al., 2017; see above). This should be described more accurately to share the precise status quo of this research field with the audience.

We thank the reviewer for the suggestion. We have adjusted and incorporated the references that correspond and explains with more detail alternative hypothesis and experimental results.

4. In Figure 2, the authors show that both the optical stimulation as well as the electrical stimulation trigger synaptic plasticity, consisting of an immediate depression, followed after a pause by a potentiation. This potentiation is – in the case of electrical stimulation – significantly different from the baseline values (not significant for the opto group, but the number of recordings is quite low). It thus is conceivable that this effect contributes to the enhanced correlation of activity in the network that is shown in Figure 1. While new experiments could better assess whether the enhanced correlation can persist with only excitability changes being available as a cellular mechanism, these are not necessary for publication in eLife. Even so, we would request that the authors discuss this potential issue in the manuscript.

We thank the reviewer for this suggestion, we have incorporated the following paragraphs in the Discussion:

The increases in correlation among neurons, generated by an increased excitability, could also contribute to generate or enhance long-term synaptic plasticity (Lisman et al., 2018; Titley et al., 2017)"

"Primary visual cortex (V1) responds to sensory information. Thus, one of the main roles of V1 neurons is to integrate sensory inputs (Buonomano and Merzenich, 1998; Mountcastle, 1997) (Figure 8). Unitary EPSP, i.e., pyramidal to pyramidal monosynaptic connections, would not be expected to increase the firing probability of postsynaptic neurons, in vitro or in vivo (Holmgren et al., 2003; Jouhanneau et al., 2018). However, the joint activation of several neurons in the local circuit (in our case: neighbour neurons with opsin expression), could evoked compound EPSC and EPSP with amplitudes larger than unitary EPSC or EPSP, and could generate synfire chains of activity that propagate through the cortex (Abeles, 1991). Thus, increases in synaptic efficacy, mediated by an increase in neuronal excitability of connected neurons, could play a significant role in the local circuit, improving the efficiency of the flow of the information during visual stimulation and during the recall of neurons associated with a memory."

Reviewer #2 (Recommendations for the authors):I believe that this is a strong paper with a lot of potential for very high impact, but it all depends on whether it can be demonstrated that in the true absence of synaptic gain changes (or with those being at least reduced) the observed ensemble effect persists -- only with intrinsic plasticity as an available cellular mechanism.This will take a bit of time and work, but it could make the paper a game-changer for plasticity research.

We agree with the reviewer that the final dissection of synaptic plasticity vs. excitability could be a game changer. We are nevertheless excited to share these results that point to a major change in excitably, with the field. We are continuing future work in vivo to better define the potential role of synaptic plasticity and evaluating its importance.